**Subject Category:**
Biology (whole organism)

behaviour/ecology

European eel, orientation, lunar cycle, migration

**Author for correspondence:**
Alessandro Cresci
e-mail: alessandro.cresci@rsmas.miami.edu

# The relationship between the moon cycle and the orientation of glass eels (*Anguilla anguilla*) at sea

Alessandro Cresci[1,2], Caroline M. Durif[2], Claire B. Paris[1], Cameron R. S. Thompson[2], Steven Shema[3], Anne Berit Skiftesvik[2] and Howard I. Browman[2]

[1]Department of Ocean Sciences, Rosenstiel School of Marine and Atmospheric Science, 4600 Rickenbacker Causeway, Miami FL 33149-1098, USA
[2]Institute of Marine Research, Austevoll Research Station, Sauganeset 16, N-5392 Storebø, Norway
[3]Grótti ehf., Grundarstíg 4, 101 Reykjavík, Iceland

AC, 0000-0001-5099-3520; CMD, 0000-0002-9405-6149;
CBP, 0000-0002-0637-1334; CRST, 0000-0003-3318-4064;
ABS, 0000-0002-7754-5661; HIB, 0000-0002-6282-7316

Links between the lunar cycle and the life cycle (migration patterns, locomotor activity, pulses in recruitment) of the European eel (*Anguilla anguilla*) are well documented. In this study, we hypothesized that the orientation of glass eels at sea is related to the lunar cycle. The European eel hatches in the Sargasso Sea and migrates across the Atlantic Ocean towards Europe. Upon reaching the continental shelf, the larvae metamorphose into glass eels and migrate up the estuaries, where some individuals colonize freshwater habitats. How glass eels navigate pelagic waters is still an open question. We tested the orientation of 203 glass eels in a transparent circular arena that was drifting *in situ* during the daytime, in the coastal Norwegian North Sea, during different lunar phases. The glass eels swimming at sea oriented towards the azimuth of the moon at new moon, when the moon rose above the horizon and was invisible but not during the other moon phases. These results suggest that glass eels could use the moon position for orientation at sea and that the detection mechanism involved is not visual. We hypothesize a possible detection mechanism based on global-scale lunar disturbances in electrical fields and discuss the implications of lunar-related orientation for the recruitment of glass eels to estuaries. This behaviour could help glass eels to reach the European coasts during their marine migration.

# 1. Introduction

The European eel (*Anguilla anguilla*) undertakes a long-distance migration of greater than 5000 km twice during its life, from the spawning areas in the Sargasso Sea to the European coast and back [1]. After hatching in the Sargasso Sea, eel leptocephalus larvae drift with the gulf stream [2,3] until they reach the continental slope of Europe. There, they metamorphose into the transparent post-larval glass eel [4], which then migrate across pelagic waters towards coastal areas [3,5]. Glass eels are attracted to estuaries and most of them will migrate upstream into freshwater habitats. There, eels will spend most of their lifetime (5–30 years) as yellow eels before becoming silver eels [3,6]. Silver eels then navigate back to the Sargasso Sea where they spawn and presumably die [7,8].

The mechanisms that glass eels use to orient towards the coast are not well understood. Eels have an extremely sensitive olfactory system [9] and glass eels are possibly attracted by chemical cues, such as freshwater plumes containing inland odours [10,11], when they swim near the coast. Additionally, glass eels sense the Earth's magnetic field and use it as a reference compass mechanism to orient [12], as do many other species [13]. Specifically, glass eels showed a magnetic swimming direction on a north–south axis, which was possibly related to the tidal flows of the stream estuary where they were collected (estuary flowing towards the north during ebb tide). However, while tidal flow can affect the swimming behaviour of glass eels in estuarine and shallow coastal areas [14], the nature of the directional stimulus that glass eels use to swim in a specific compass direction in open pelagic waters is unknown. Orientation at sea could be based on magnetic cues alone, which would provide glass eels with a perceptual mechanism to maintain an innate compass direction. However, in the case of glass eels, orientation at sea does not follow a direct trajectory, and changes in orientation behaviour occur in association with the tidal cycle [12]. Thus, the orientation mechanism involved at this stage of their migration might be more complex and multifaceted, and the magnetic sense could also serve as a stable frame of reference for the interpretation of other directional cues (e.g. salinity gradients, odour plumes and water currents).

The movement ecology and behaviour of many species of anguillids, from the larval to the adult stage, is related to the phase of the moon. For example, the spawning of Japanese eel (*Anguilla japonica*) is synchronized with the new moon cycle [15]. Leptocephalus larvae of Atlantic eels (*A. Anguilla* and *Anguilla rostrata*) change their depth distribution in the open ocean according to the moon cycle, swimming deeper during full moon [16]. The arrival of glass eels at the coast and estuaries is related to the moon cycle [17]. Pacific glass eels (*A. japonica* and *Anguilla marmorata*) show the highest recruitment to coastal areas at new moon, and *Anguilla bicolor pacifica* recruits mostly during full moon [18]. The arrival of glass eels of *Anguilla australis* and *Anguilla dieffenbachii* to the New Zealand coast is linked to the moon cycle, with the highest number of eels observed during new moon and full moon [19]. This pattern was observed at several locations in New Zealand [20]. There is also evidence that the behaviour of adults is linked to the moon cycle [21,22]. Adult yellow eels tagged in the Mediterranean Sea swim slower and closer to the bottom during full moon [23]. Interestingly, freshwater adult silver eels migrate downstream towards the sea mostly around new and full moon, independent of the percentage of the moon visible [22,24,25]. We hypothesized that the moon could also provide glass eels with a compass direction at sea that could guide them in reaching the coast.

To investigate the influence of the moon on the orientation of European glass eels, we observed the orientation behaviour of 203 glass eels *in situ* during the daytime, while drifting in a transparent circular behaviour arena, the drifting *in situ* chamber (DISC, figure 1) in the coastal Norwegian North Sea (figure 2) during the four main moon phases: full moon, third quarter, new moon and first quarter. Additionally, we compared glass eel orientation to the position of the moon relative to the horizon. The data collected were used to assess the relationship between the orientation direction of the glass eels and the moon azimuth (the angle between the north and the vertical projection of the moon onto the horizon). Furthermore, the influence of multiple lunar-related factors, such as moon phase, moon position relative to the horizon (above/below) and tide (ebb/flood) on the orientation and swimming speed of glass eels was explored.

# 2. Results

The experiments were designed to investigate whether the moon influences the orientation of glass eels throughout the lunar cycle. Glass eels (150) were tested individually while drifting in the DISC in the

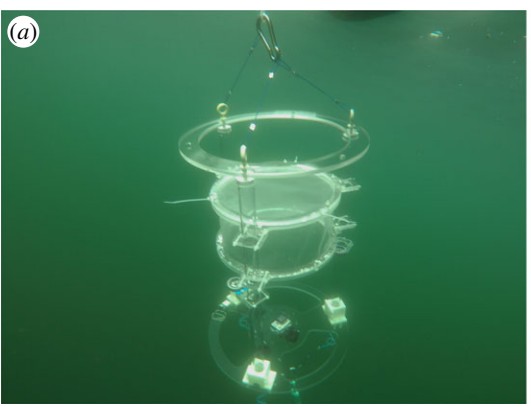
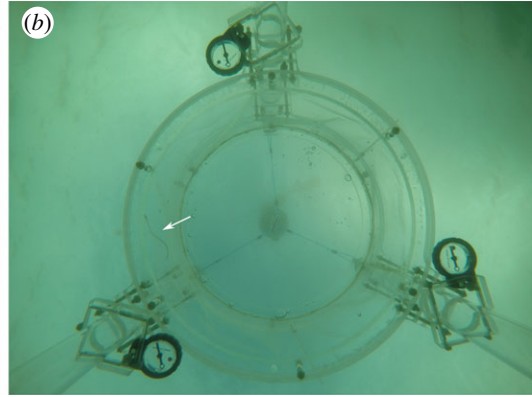

**Figure 1.** Drifting *in situ* chamber (DISC) used to observe glass eels (*Anguilla anguilla*) *in situ*. (*a*) View of the DISC drifting in the North Sea. The acrylic frame, the circular behavioural arena and the GOPRO camera placed underneath are visible in the picture. (*b*) View from the GOPRO camera underneath the arena. The white arrow points towards a glass eel swimming in the arena. Below the arena, attached to the poles, three analogue magnetic compasses which were used, together with a digital compass, to monitor the orientation of the glass eels with respect to the magnetic north.

North Sea. The dataset also includes the orientation data of 53 glass eels tested under the same conditions published in Cresci *et al.* [12]. Altogether, the dataset includes 203 individual glass eels observed at sea (table 1). DISC deployments were conducted during the four main moon phases: full moon, third quarter, new moon and first quarter (table 1).

For each glass eel, we calculated the mean orientation direction (mean bearing) and its significance based on video recordings (details of the data collection and statistical analysis are presented in electronic supplementary material, figure S1). Of the 203 glass eels observed, 175 (86%) showed non-random orientation and a significant mean bearing (Rayleigh's $p < 0.05$). This proportion of eels showing orientation was highest during new moon and first quarter (96% and 93%, respectively) was lower during full moon (80% of the eels oriented) and was lowest during the third quarter (62%). The associated mean $R$ values (Rayleigh test, ranging from 0 to 1) were $0.30 \pm 0.16$ and $0.30 \pm 0.13$ (mean $\pm$ s.d.) during new moon and first quarter, $0.26 \pm 0.14$ during full moon and $0.14 \pm 0.06$ during the third quarter.

We compared the mean individual orientations of the 175 significantly orienting glass eels between the different tidal and lunar configurations (table 2). Concerning the orientation directions, we considered three variables: the orientation direction of every glass eel with respect to the magnetic north ($\alpha_{north}$), the azimuth of the moon and the angle between the orientation of the eel and the azimuth of the moon ($\alpha_{moon}$) (figure 3). Only glass eels tested during the new moon and first quarter showed a common orientation towards the moon azimuth (figure 4*c*,*d*, column iii). During the ebb tide at new moon, glass eels oriented towards the moon azimuth (figure 4*c*, column iii), the moon azimuth was south (figure 4*c*, column ii) and glass eels oriented to the south (figure 4*c*, column i). During the flood tide at new moon (figure 4*c*), tests were performed while the moon azimuths were in opposite directions (East–West) (figure 4*c*, column ii). Under these conditions, the behaviour of glass eels did not change; they still oriented towards the azimuth of the moon (figure 4*c*, column iii). However, there was no common orientation towards the magnetic south (figure 4*c*, column i).

The orientation towards the new moon azimuth was independent of the tidal phase. When all of the data collected during the new moon were combined, there was a highly significant orientation towards the azimuth of the moon [$N = 80$, Ray. mean bearing = 11.2°, $r = 0.38$, $p = 0.000006$; figure 5*a*(iii)] and a significant orientation to the magnetic south [$N = 80$, mean bearing = 180°, $r = 0.22$, $p = 0.02$; figure 5*a*(i)].

During the first quarter (figure 4*d*), when the percentage of moon visible to the eye increases, the moon azimuth was towards the northeast and a common orientation towards the southeast was observed (figure 4*d*, column i), along with a significant for orientation with respect to the moon azimuth (figure 4*d*, column iii), although only when the moon was above the horizon (occurring only during flood tide) (figure 4*d*, column ii). The common orientation (figure 4*d*, column i and iii) towards magnetic south and the moon azimuth observed during flood tide disappeared during the ebb tide, when the moon fell below the horizon (figure 4*d*, column ii). No significant patterns in orientation direction were observed during full moon, when the moon was always below the horizon (figure 4*a*), and third quarter (figure 4*b*).

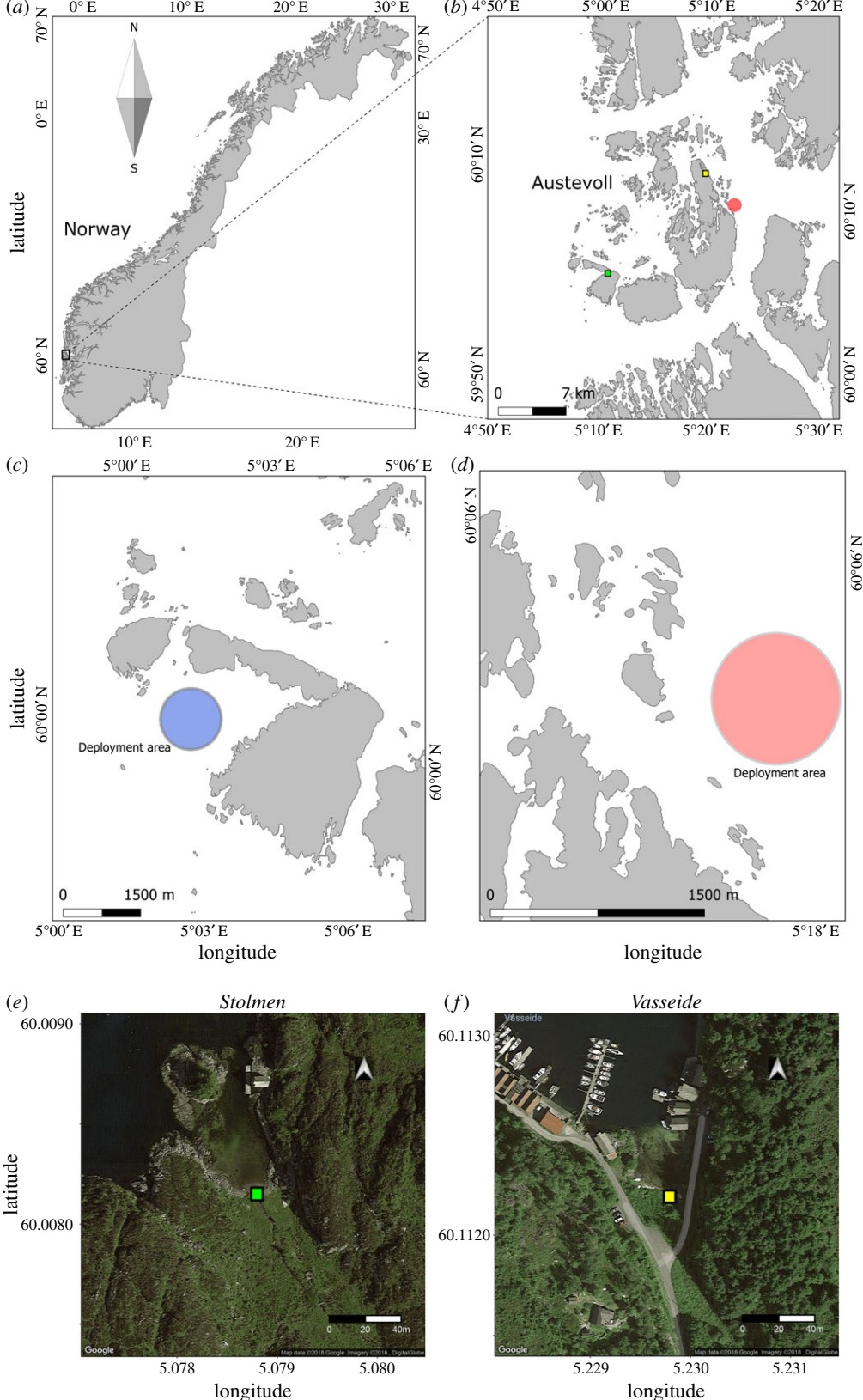

**Figure 2.** Map of the locations where glass eels (*Anguilla anguilla*) were observed drifting in the DISC *in situ*. (*a*) map of Norway. (*b*) Map of the archipelago of Austevoll. (*c,d*) Maps of the deployment areas of Stolmen (blue) and Langenuen fjord (red). (*e,f*) Satellite view of the estuaries of Stolmen (green box) and Vasseide (yellow box).

**Table 1.** Experimental conditions during the drifting *in situ* chamber (DISC) deployments to test the effect of the moon phase on the orientation of glass eels (*Anguilla anguilla*). N, the number of glass eels.

| moon phase | N | period | percentage of moon visible (%) |
|---|---|---|---|
| full moon | 57 | 11–16 Apr 2017 | 80–99 |
| third quarter | 32 | 19–20 Apr 2017 | 42–52 |
| new moon | 53 | 17–21 Apr 2015 | 0–10 |
| | 30 | 27 Apr 2017 | |
| first quarter | 31 | 3–4 May 2017 | 57–67 |

**Table 2.** Summary statistics of the orientation of glass eels (*Anguilla anguilla*) according to magnetic north, tidal and moon phase and the moon's position at the horizon. N, number of eels. Mean bearings represent the topographical bearing of the eel relative to magnetic north or to the moon azimuth (0° = toward the moon, 180° = away from the moon). These are only presented when significant (Rayleigh test, r is the angular dispersion, p = Rayleigh's p-value). The moon position either below or above the horizon is provided for each test configuration. Fig., reference figure where those data are displayed.

| moon phase | tidal phase | moon at horizon | N | magnetic north mean bearing (°) $\alpha_{north}$ | r | p | moon azimuth mean bearing (°) $\alpha_{moon}$ | r | p | fig. |
|---|---|---|---|---|---|---|---|---|---|---|
| full | flood | below | 20 | — | 0.09 | 0.86 | — | 0.05 | 0.96 | 4a |
| | ebb | below | 26 | — | 0.17 | 0.48 | — | 0.29 | 0.10 | 4a |
| third q. | flood | below | 11 | — | 0.09 | 0.92 | — | 0.06 | 0.97 | 4b |
| | ebb | above | 9 | — | 0.48 | 0.12 | — | 0.49 | 0.11 | 4b |
| new | flood | above | 35 | — | 0.17 | 0.37 | 3° | 0.31 | 0.04 | 4c |
| | ebb | above | 45 | 195° | 0.42 | 0.0002 | 16° | 0.43 | 0.0001 | 4c |
| first q. | flood | above | 13 | 147° | 0.64 | 0.003 | 75° | 0.68 | 0.0014 | 4d |
| | ebb | below | 16 | — | 0.32 | 0.18 | — | 0.35 | | 4d |

Since the tests were conducted during the day, we also analysed the orientation data according to the sun azimuth and the light intensity during the tests (electronic supplementary material, §2, figures S5–S8). Neither had a significant relationship to the orientation of the glass eels (electronic supplementary material, §2).

The swimming speed of glass eels was significantly faster when the moon was above the horizon (median = 3.0 cm s$^{-1}$) compared to when the moon fell below the horizon (median = 2.6 cm s$^{-1}$) (Wilcoxon test p = 0.0002) (electronic supplementary material, figure S3).

## 3. Discussion

Glass eels significantly oriented towards the azimuth of the moon at sea only during specific phases of the lunar cycle. Specifically, a significant orientation with respect to the moon azimuth was observed at new moon, when the moon was above the horizon. With these conditions, the visibility of the moon is minimal (at this time, the moon is invisible). The clearest response was observed during new moon/ebb tide, when glass eels also oriented to the south, but it was different during new moon/flood tide, when the azimuth of the moon was split between east and west. Under these conditions, glass eels oriented towards the moon azimuth but did not orient with respect to the north.

Significant patterns in orientation direction were also observed during the first quarter, but only during flood tide, when the moon was above the line of the horizon. However, the orientation response was 75° from the moon azimuth and eels oriented to the southeast. This orientation was lost during ebb tide when the moon was below the horizon. During full moon and third quarter, we did not observe patterns in

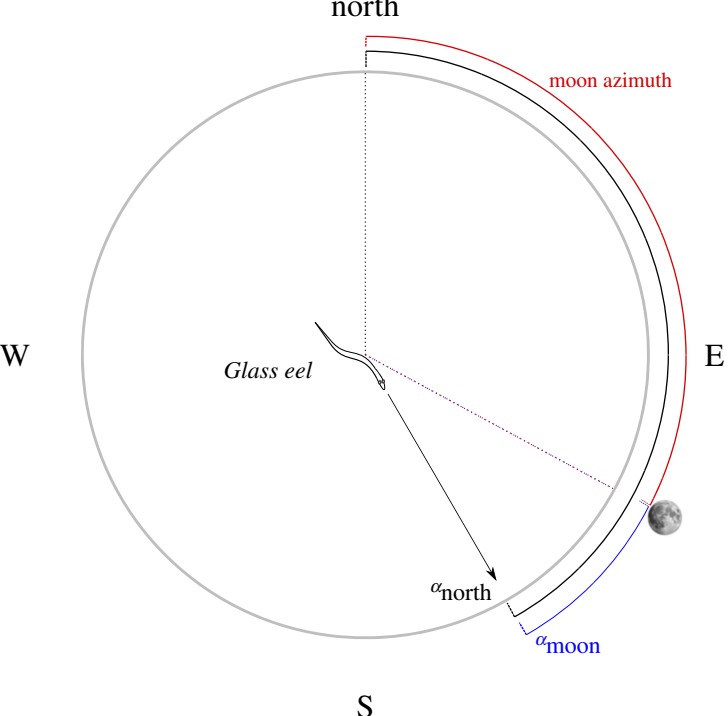

**Figure 3.** Diagram of the circular variables considered in this study. At the centre of the circle, a glass eel (*Anguilla anguilla*) swimming at sea on the horizontal plane, with respect to the Earth's magnetic north, east, south and west. The black arrow shows the orientation direction of the eel ($\alpha_{north}$), which is the angle between the mean bearing of the eel and the magnetic north (black arch). The moon azimuth is the red arch, which is the angle between the magnetic north and the orthogonal projection of the moon onto the horizon. The blue arch is the angle between the orientation direction of the eel and the moon azimuth ($\alpha_{moon}$) is the angle between the orientation of the eel and the moon azimuth.

orientation behaviour. Compared to the new moon phase, the features of the lunar cycle reverse during full moon. At new moon, the moon is above the horizon mostly during the day (and below the horizon during the night), while at full moon, the moon is above the horizon during the night. Such features of the lunar cycle might explain the difference in behaviour observed in this study.

Several biological processes are linked to the phases of the moon. Diel vertical migration of zooplankton is entrained in the lunar cycle during the long Arctic winter [26]. Mass spawning of corals occurs during certain full moon and last-quarter nights of the year [27]. Similar correlations were observed in shrimps: growth rate and moulting of the penaeid shrimp *Penaeus vannamel* coincides with the lunar phase [28]. The lunar cycle is also correlated with the spawning of many fish. For example, Nassau grouper (*Epinephelus striatus*), Atlantic killifish (*Fundulus heteroclitus*), salmonids and other reef fishes all spawn during particular lunar phases [29–32].

The orientation of glass eels in our study was related to the moon's azimuth and its position relative to the horizon, suggesting that the moon might serve as a possible additional cue during their migration towards land. This moon-related orientation could play a role during the migration of glass eels across the continental shelf, in addition to chemical and tidal cues and/or when those cues are missing or unreliable. Glass eels are known to use odours [11,33,34], salinity gradients [35] and selective tidal stream transport (STST) [36–39] in coastal and brackish water. However, these cues could be absent or less reliable in pelagic areas or far from large freshwater inputs coming from land. Moon-related orientation might work together with the magnetic compass during the pelagic step of the migration over the continental shelf. One possibility is that the moon might serve as a directional stimulus and the magnetic compass as a frame of reference. Another possibility is that the two systems might alternate such that when the moon is undetectable eels might use the compass to maintain a course learned during previous new moon phases. These hypotheses will be tested in future research.

Glass eels oriented towards the moon azimuth only at new moon (when the moon was always above the horizon), and partially during the first quarter but with lower precision (figure 4). However, during the first quarter, this relationship disappeared when the moon was below the horizon. Glass eels also swam at a significantly higher speed when the moon was above the horizon. These results suggest that glass eels

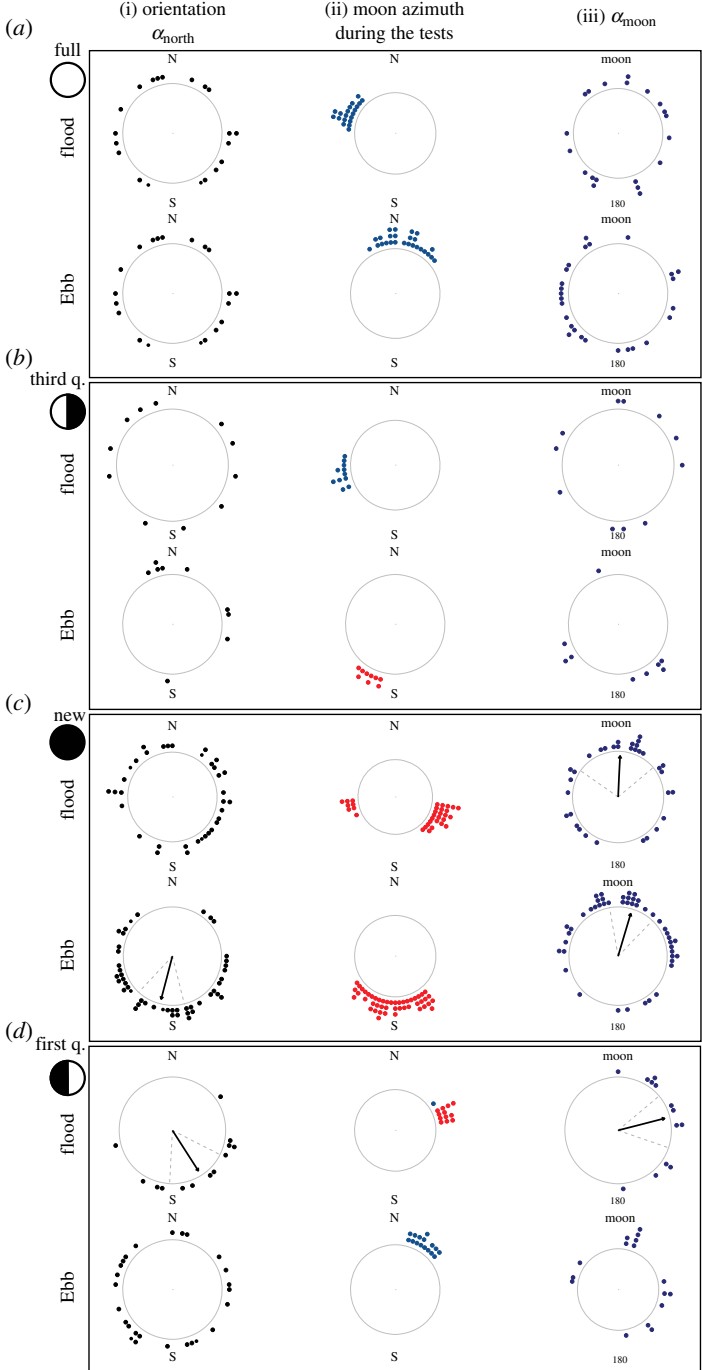

**Figure 4.** Orientation of glass eels (*Anguilla anguilla*) at sea and moon-related orientation. Orientation of the glass eels, the azimuth of the moon and orientation with respect to the azimuth of the moon (as described in figure 3), during each of the four main moon phases [(*a*): full moon, (*b*): third quarter, (*c*): new moon, (*d*): first quarter] and during each tidal phase (flood/ebb). The frame of reference of each plot is indicated: N = magnetic north, MOON = direction of the moon at the horizon (azimuth). In the first and third column, significant preferences of orientation direction are shown by a black arrow starting from the centre of the circle and pointing towards the mean orientation direction. Dashed grey lines are the 95% confidence intervals around the mean. The circular plots are empty when there was no significant preference of orientation direction. The first column i. Orientation ($\alpha_{north}$) shows the orientation of glass eels with respect to the magnetic north (0°) and south. Each black data point represents the mean bearing of a single glass eel. Only the glass eels that had a significant individual orientation are presented. The second column, ii. Moon azimuth during the tests, shows the direction of the moon azimuth during each DISC deployment. The points are blue if the moon was below the horizon and red if the moon was above the horizon. The third column, iii. ($\alpha_{moon}$), shows how far the glass eels were orienting from the moon azimuth (MOON, top of the plot = 0°). The angular difference between the orientation of each eel and the azimuth of the moon is shown as a navy-blue data point. Significant collective orientation towards the direction of the moon is shown as a black arrow pointing towards the top of the plot (at new moon and first quarter).

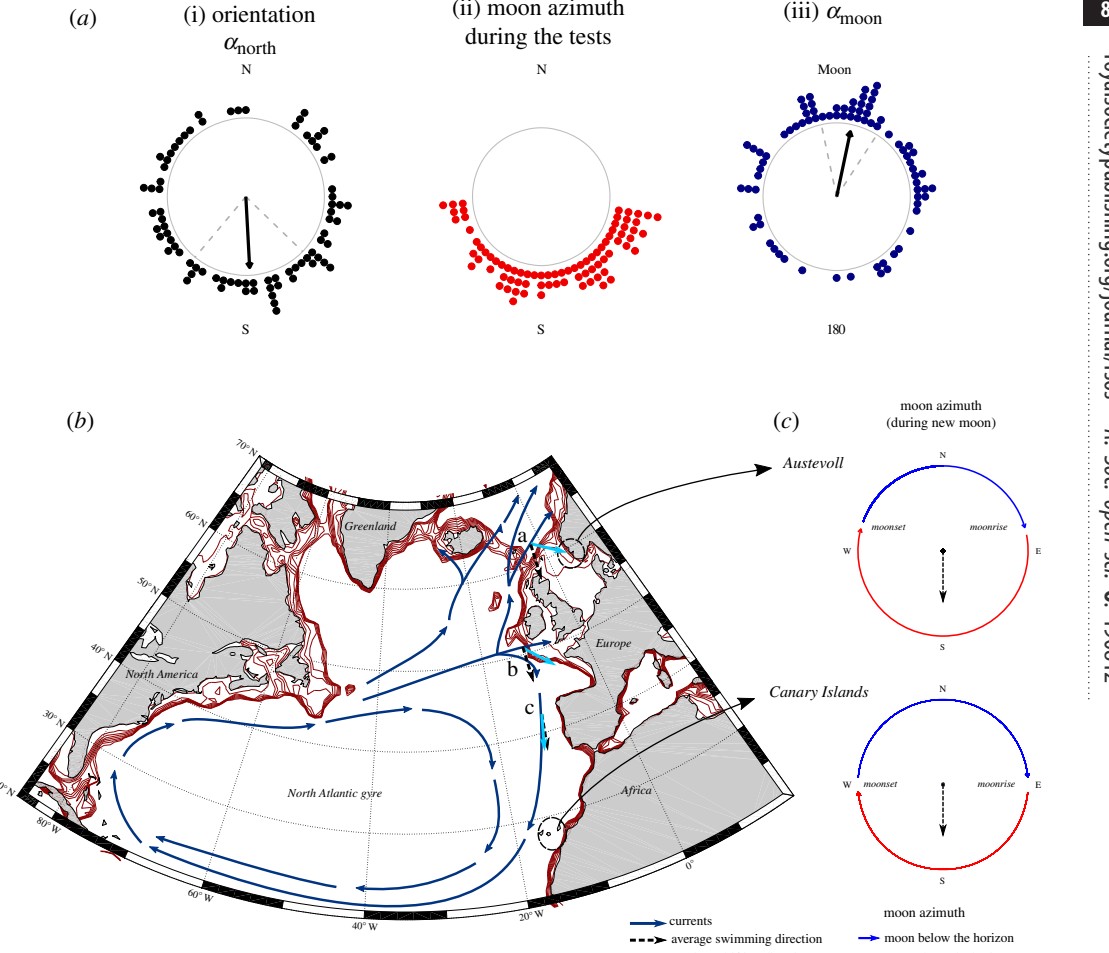

**Figure 5.** Schematic illustration of the potential advantages of the moon-driven orientation during the migration of European glass eels (*Anguilla anguilla*). (*a*) Pooled data of collective orientation ($\alpha_{north}$) (i), moon azimuth during the tests (ii) and orientation with respect to the moon azimuth ($\alpha_{moon}$) (iii) at *new moon*. The circular plots have the same features as described in figure 5. (*b*) Map of the North Atlantic Ocean with a simplified schematic representation of the great North Atlantic gyre and the main surface currents in proximity of the European continental shelf (navy-blue arrows). The continental shelf is visible from bathymetric red lines (0–700 m, with 100 m interval between the lines). (*c*) simplified plots of the moon path at the horizon at Austevoll, Norway (where the experiments were conducted and close to the northern limit of the area of distribution of glass eels), and at the Canary Islands, West of Africa (southernmost point where glass eels are found) during the new moon periods of March–May 2017. The plots show approximate moonrise and moonset with respect to the magnetic north (with around 10° of possible variation). Red arrows show the moon path at the horizon when the moon is above the line of the horizon, and the blue arrows when the moon is below the line of the horizon. Dashed black arrows in (*b,c*) show the average swimming direction that the glass eels would have swimming towards the direction of the moon when the moon is above the horizon, at new moon (south, as showed by the *in situ* orientation data displayed in (*a*)). In (*b*), we report three possible case scenarios of the drifting direction (sky-blue arrows) resulting from the passive drifting caused by the currents and the moon-driven orientation (south oriented). (*b*.a) shows that swimming towards the moon could potentially help glass eels to exit the Norwegian Current, enter the North Sea and arrive at the coast. (*b*.b) shows that the moon-oriented swimming could help the glass eels coming from the Gulf Stream to arrive at the Bay of Biscay, the area with the highest recruitment of glass eels. (*b*.c) shows that the moon-oriented swimming could help glass eels to reach also the Canary Island at the southernmost point of their distribution.

only detect the position of the moon when it is above the horizon at new moon. However, the presence of the moon above the horizon is important but it is not the only factor that plays a role. When the moon is above the horizon, the moon phase also matters. The orientation behaviour of the glass eels changed according to the three moon phases during which the moon was above the horizon: third quarter, new moon and first quarter. During the third quarter, despite the moon being above the horizon during half of the tests, the glass eels did not follow the direction of the moon. Similarly, during the first quarter, glass eels had a common orientation direction when the moon was above the horizon, but they did not orient towards the moon azimuth with the same precision as observed during new moon.

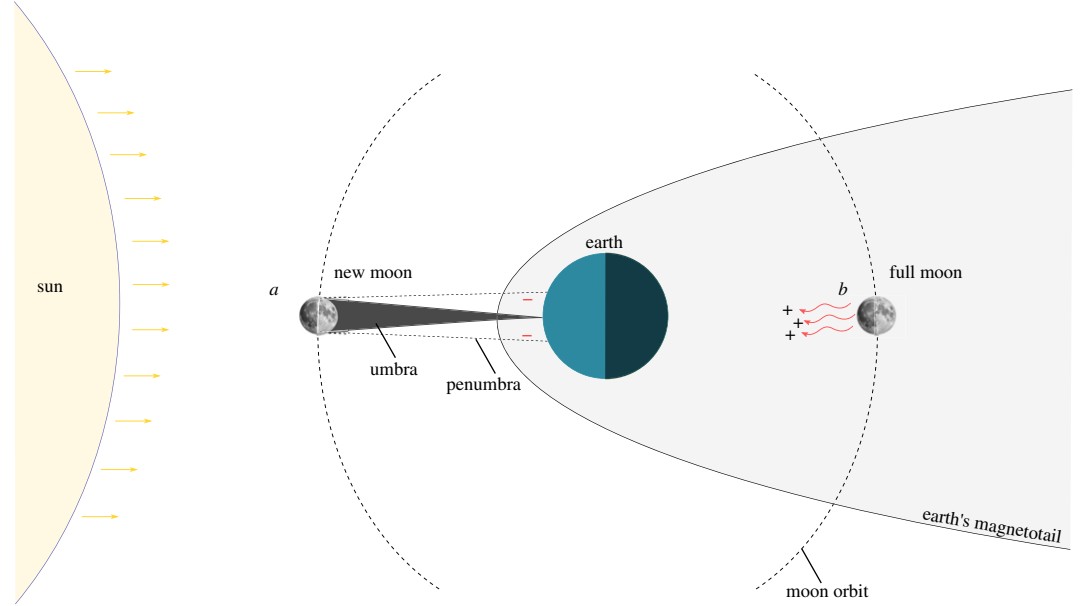

**Figure 6.** Diagram of the electrical disturbance of the moon to the Earth's surface (modified from: Bevington, [40]). (*a*): at new moon, the moon faces the bright side of the Earth (moon above the horizon during the daytime) and the solar wind impacts the moon. This creates an umbra and a penumbra, which are shadows of the moon body carrying negative electric charges to the Earth's surface. (*b*) at full moon, the moon faces the dark side of the Earth (moon above the horizon at nighttime). Following its orbit, the moon impacts the Earth's magnetotail and creates an electric disturbance of positive charges propagating towards the Earth's surface.

## 3.1. Hypothetical mechanism for the detection of the moon azimuth

The behaviour of the glass eels showed the strongest lunar-related patterns in orientation only when the moon was invisible to them (at new moon during the daytime). Therefore, the mechanism involved in the detection of the moon azimuth during the day cannot be visual. The moon's gravitational pull (1) and the change in electric flux caused by the moon (2) represent two other possible mechanisms explaining glass eel orientation towards the moon azimuth.

The gravitational pull of the moon has significant effects on large masses, such as water in the oceans. It has a much smaller effect on small masses, such as planktonic organisms. The gravitational pull of the moon on a person weighing 80 kg is 0.0027 N, which is $10^5$ lower than the gravity of the Earth (785 N) [40]. Thus, although theoretically possible, observable biological effects of the moon phase on the glass eel behaviour are probably not due to the lunar gravitational pull.

The moon generates electrical disturbances at the level of the Earth's surface, which depend on the relative positions of the moon, the Earth and the sun [40] (figure 6). During full moon (figure 6*b*), the moon faces the dark side of the Earth (moon above the horizon only at night). During this phase, the Earth is between the moon and the sun. Consequently, the motion on its own orbit causes the moon to cross the Earth's magnetotail (figure 6*b*) and, in doing so, disturbs it. The Earth's magnetotail contains an earthward-directed electric field [41], and when the moon passes through it, there is an ion exchange between the moon's surface and the Earth's magnetotail [42]. This phenomenon causes variations in the electric field at the level of the Earth's surface. Measurements of electric fields during the lunar cycle revealed that, during full moon nights, there is a positive voltage increase from less than or equal to $1 \text{ V m}^{-1}$ (baseline value occurring during the daytime, when the moon is still below the horizon) to $1–2 \text{ V m}^{-1}$, with peaks up to $16 \text{ V m}^{-1}$ [40]. A similar electric disturbance also occurs at new moon, but it is caused by a different astronomical process [40,42]. At new moon, the moon is between the Earth and the sun (figure 6*a*). The solar radiation (solar wind) impacts the surface of the moon and creates a lunar wake that propagates downstream towards the Earth (figure 6*a*). The obstruction of the solar wind causes an accumulation of negative electric charges on the side of the moon facing the Earth (anti-solar side) [42]. Thus, the moon becomes an 'electric dipole' propagating a negatively charged electrical field downstream [42] to the Earth's surface during the daytime. These electrical phenomena potentially influence the behaviour of many electrosensitive animals both on land and in the aquatic environment [40,43–45].

Clear examples of biological effects of weak electric fields comparable to those caused by the moon come from spiders, which use atmospheric electricity to take off from the ground and fly hundreds of kilometres, a phenomenon known as 'ballooning' [43]. These electric fields could also affect marine animals, as seawater is a conductive medium [40]. This could also explain why biological phenomena, such as spawning of multiple deep-sea species, are correlated with the lunar cycle [44], at depths to which the moonlight penetrates at very low intensities during dark nights, and only in clear water [46]. Interestingly, the glass eels observed in this study showed a preferred orientation direction towards the azimuth of the moon during the daytime at new moon, which is exactly when the electrical field of the new moon impacts the Earth's surface.

Eels are among the taxa of marine organisms that are electrosensitive. Previous work investigating the effect of weak electric fields (less than $1\ \text{V}\ \text{m}^{-1}$) on the orientation of juvenile American eels (*A. rostrata*) revealed that, when swimming in a square arena under artificial electric fields, eels turned towards the anode (negative charges), and that reversing the polarity of the electric field inverted their turning direction [47]. The authors also suggested that eels could use electric fields present in the ocean for orientation. Similar experiments were conducted on the European eel (using adult eels), although such high sensitivity was not observed [48]. The results from Zimmerman and McCleave [47] are consistent with the hypothesis that glass eel orientation towards the moon azimuth could possibly involve the perception of lunar cycle-related electric fields.

The lack, or partial lack, of moon-related orientation during the third quarter and first quarter (figure 4b,d) is also consistent with the electric field hypothesis because during these lunar phases, there are no electric disturbances of the moon to the Earth's surface [40]. In future work, we intend to investigate whether glass eels orient towards the moon during full moon nights using infrared cameras, and conduct experiments on glass eels' sensitivity to electric fields comparable to those caused by the moon.

## 3.2. How orienting towards the moon could help glass eels during the pelagic landward migration

Following the azimuth of the moon at new moon could play a role in the recruitment of the European eel across Europe. The European eel constitutes a single panmictic population [49,50] that inhabits marine and freshwater habitats from northern Norway to Morocco [3]. The entire population spawns in the same area in the Sargasso Sea [1,51]. This means that the European eel must possess a generic orientation mechanism to reach the continent while also supporting latitudinal spread across Europe. At new moon in Austevoll, Norway (where the tests were conducted), during the period when glass eels arrive at the coast (March–May), moonrise is approximately eastward (figure 5c). The moon then moves on its path above the horizon all the way southward and sets below the horizon approximately westward (figure 5c). This means that swimming towards the moon azimuth when the moon is above the horizon at new moon would lead to an average swimming direction towards the south (figure 5c). This average southward orientation direction at new moon is consistent with our observations: glass eels significantly oriented towards the moon azimuth and this corresponded to a southward direction (figure 5a). At a larger scale, the direction of the moonrise and the moonset at new moon during the period of arrival of glass eels does not change much with latitude. In the Canary Islands (southernmost point of the distribution of the European eel) [52,53], the moon still rises eastward and sets westward (figure 5c). Thus, the average orientation direction of glass eels would always be south during this moon phase, independent of latitude (figure 5b,c). Additionally, during full moon at night, the moon has the same path compared to that during new moon; it rises east, it moves southwards and it sets in the west. Thus, if glass eels have similar moon-driven orientation during full moon nights (when the electrical disturbance of the moon is the strongest, figure 5b), they would swim southward for approximately 8 days a month (approx. 4 days of new moon, approx. 4 days of full moon), when the moon is above the horizon. This is consistent with the timing of the recruitment of many species of glass eels, which peaks both at new and full moon [17–19] when the strongest tidal flows also occur.

Following from the above, it is possible to develop several working hypotheses about recruitment of glass eels to the European coasts (figure 5b). Glass eels metamorphosing on the continental slope at the northern edge of the North Sea (figure 5a, location a) need to actively swim towards the south to exit the Norwegian Atlantic Slope Current (NwASC) (which otherwise would transport them to the Arctic) and enter the North Sea. The NwASC flows between the Faroe Island and Scotland and travels northward

following the North Sea and the Norwegian continental slope, at an average speed above 30 cm s$^{-1}$ [54]. From our *in situ* data, we observed that glass eels were swimming at $3.02 \pm 0.68$ cm s$^{-1}$ (mean $\pm$ s.d.). Laboratory measurements on the critical speed (Ucrit) of glass eels indicate higher values of 11–13 cm s$^{-1}$ [55]. Swimming to the south, towards the moon azimuth, would play an important role especially at the boundary areas of the NwASC, where exiting the current or drifting with it could make the difference between entering the North Sea and successfully recruiting to the coast, or drifting to the Arctic.

In the central part of the species' distribution (figure 5*b*, location b), orienting towards the south could facilitate migration over the continental Celtic Shelf and the Porcupine basin, and towards the Bay of Biscay (figure 5*a*, location b). This bay is a critical recruitment area for the European glass eel and it is where the largest glass eel fishery in Europe takes place (87% of all European glass eel fished) [56].

Further south, glass eels drifting offshore off the coast of Portugal would encounter the Portugal Coastal Current (PCC) [57]. Here, a wide (1000 km) current flows steadily at approximately 10–30 cm s$^{-1}$, year-round, towards the south [58,59]. Under these oceanographic conditions, swimming towards the moon could shorten the duration of the journey of the glass eels arriving from more northern areas and migrating to the Canary Islands. However, it is important to consider that over the latitudinal range of the European glass eel distribution, the altitude (zenith) of the moon significantly increases at lower latitudes, making the moon a less horizontal directional cue at lower latitudes. Thus, moving more south than the Canary Islands, the moon would reach altitudes over 80° above the horizon, and much of its directionality would be lost.

However, it is also possible that glass eels might display differences in lunar-related orientation behaviour depending on the recruitment area. We will test these scenarios and hypotheses in future work.

# 4. Material and methods

## 4.1. Animals and maintenance

Glass eels were collected using hand nets at several estuarine streams located around the Austevoll archipelago (Norway) (figure 2), before they migrated into freshwater. The eels were found under rocks and sediment, at low tide, and the majority of them were collected from the stream estuaries of Stolmen (60.0082 N and 5.0788 E, figure 2*e*) and Vasseide (60.1122 N and 5.2298 E, figure 2*f*). The eels were collected during the recruitment period at the estuarine streams, between March and April of 2015 and 2017. None of the animals used in this study were pigmented (developmental stage: V–VI [3]), and they did not have food in the gut.

Glass eels were kept in 20 l maintenance tanks, where they were re-acclimated to near full salinity seawater (32 ppt) after capture. They were kept in aerated aquaria in a temperature-controlled room set to ambient conditions similar to those that the glass eels encountered when they arrived at the coast and would experience during the *in situ* deployments of the DISC (ranging between 6 and 10°C). Animals were not fed (they were at the pre-feeding stage) and were kept in 14 h light and 10 h dark cycle (following the daylength at the study location during the observation period). Two-thirds of the volume of each aquarium was replaced with filtered seawater every 48 h to maintain water quality. The seawater was provided by the filtering system at the Institute of Marine Research's Austevoll Research Station, which collects seawater from the Langenuen fjord at a depth of 160 m. Before being used for the deployments at sea, glass eels were taken from the large aquaria and placed in individual 500 ml white plastic containers filled with seawater at the same temperature as the aquaria. These cups were kept with the lid on but non-sealed and in a cooler to maintain temperature during transportation to the deployment sites (figure 2).

## 4.2. *In situ* observations

For all of the tests conducted *in situ*, we used the drifting *in situ* chambers (DISCs) [60,61] (figure 1), a drifting transparent circular arena that was deployed in Norwegian coastal areas near where the eels were collected (figure 2). The DISC has an acrylic structure including a circular chamber, transparent to both small-scale water movement and light. A drogue connected to the bottom of the acrylic frame holding the chamber allows the DISC to drift with the current. A fine braided line attaches the top of

the acrylic frame to a surface float, which allows easy recovery and re-deployment of the DISC. The chamber in which individual glass eels swam one at a time was 40 cm wide (diameter) and 15 cm deep. The chamber was semi open, as the bottom was rigid and made of acrylic, while the walls and the top were made of transparent fine mesh. The mesh is preferred to a rigid acrylic wall because it allows water and dissolved gas exchange, assuring that the fish in the arena could detect potential chemical cues and that the dissolved oxygen level would not decrease during deployments.

The behaviour of glass eels in the DISC was observed using a GOPRO HERO 4 camera. The device is also equipped with a HOBO Pendant® Temperature/Light 64 K Data Logger—UA-002-64, a GPS locosys gw-60, three analogue compasses and a Star-Oddi DST Magnetic digital compass. The GOPRO camera records the behaviour of glass eels viewed from underneath the chamber, looking towards the water surface (figure 1b). The digital compass is placed on the bottom plate of the frame of the DISC, oriented on the same axis as the camera. The analogue compasses are attached to the acrylic poles of the DISC frame and placed below the circular arena (figure 1b). This positioning eliminates the possibility that the compasses are a visual reference for the eel.

The Orientation With No Frame of Reference (OWNFOR) [61] approach was applied to characterize the orientation of the glass eels in the apparatus while it was drifting.

## 4.3. Deployments *in situ*

The experiments were designed to investigate whether glass eel orientation was related to the lunar cycle. Thus, DISC deployments were conducted during the four main moon phases: full moon, third quarter, new moon and first quarter. Data on moon position, percentage of moon visible and moon phase were obtained from the Norwegian calculator www.timeanddate.com/moon (Copyright© Time and Date AS 1995–2018). Data about the tide were obtained from the Norwegian Mapping Authority (www.kartverket.no). All of the deployments were conducted during the daytime.

The purpose of each trial with the DISC was to observe the orientation of glass eels, one at a time. Each glass eel was placed in the circular arena while the DISC was held semi-submerged along the side of a small boat. Once the glass eel was placed in the chamber, the lid was closed and secured using soft plastic tubes. The DISC was then gently released until it reached the depth at which it would drift throughout the test. The depth at which the glass eel drifted ranged between 4 and 5 m for all of the experiments, which is consistent with the depth range at which glass eels migrate at sea [37]. Each animal was video recorded for 15 min, at the end of which the DISC was recovered and the glass eel was replaced with a new one. The first 5 min were considered as an acclimation period; the orientation behaviour of glass eels was observed during the last 10 min of each trial [12,60].

The DISC was deployed in the fjords of Langenuen (northeast of Austevoll, 60.09 N, 5.28 E; saltwater) and in Stolmen (southwest of Austevoll, 60.00 N, 5.04 E) (figure 2), where it drifted in water that was 70–100 m deep, 300–500 m from the coast, at water temperatures ranging between 6 and 10°C.

## 4.4. Data analysis

The orientation of glass eels was determined through the analysis of the GOPRO images, tracking the position of the head of the eel in the circular arena every second for 10 min (a schematic diagram of all the steps of the circular analysis is in electronic supplementary material, figure S1). The DISC was allowed to rotate, and the position of the eels with respect to the Earth's magnetic north was monitored using the digital compasses. The video frames were processed using the DISCR tracking procedure, using R and a graphical user interface provided by imageJ software [62,63]. Using this tracking procedure, we collected the positional data (in units of magnetic degrees) of the glass eel with respect to the centre of the chamber, which were considered as bearings. The images were geo-referenced with respect to the geomagnetic cardinal points, against the reference of the digital compass. The code used is available at the Web page Drifting In Situ Chamber User Software in R (https://github.com/jiho/discr written by Jean-Olivier Irisson (Université Pierre et Marie Curie UPMC), released under the GNU General Public License v. 3.0.

Data analysis consisted of two steps. First, the mean orientation of each individual was computed from the bearings collected by the video tracking analysis. The mean of 600 data points, which represent the bearing of the fish in the drifting chamber at each second (one position/s over 10 min period), as the orientation of each individual [60–62,64]. Each bearing of the eel was corrected with respect to the magnetic north using the digital compass. At the end of this procedure, we used the

corrected tracks to compute the magnetic bearing of the glass eel, defined as the direction of motion of the fish (in degrees) with respect to the magnetic north.

The ability of each individual to keep a specific magnetic bearing while swimming in the DISC was considered evidence of directionality [61,62]. The significance of the directionality was assessed using the Rayleigh test of uniformity for circular data, and the level of convergence of the bearings towards one direction using the Rayleigh test r value (from 0 to 1) [62,63,65]. The Rayleigh's $r$ values indicate the concentration of the positions of the fish in a specific section of the circular arena, or, in other words, the accuracy of the directionality of the swimming behaviour observed. An outcome was considered statistically significant when $p < 0.05$ ($\alpha = 0.05$) [62].

After assessing the orientation of each individual, the next step of the analysis focused on evaluating whether the eels tested in the DISC have a common, collective trend in orientation (i.e. whether they go towards a common direction). To accomplish this step of the analysis, we applied the Rayleigh test of uniformity to the values of all the mean individual bearings, testing whether the frequency distribution of the directions displayed by the individuals was significantly different from random (95% confidence interval, $\alpha = 0.05$) [62].

The swimming speed of the eels was calculated from the video tracks, dividing the distance that the animal swam at every frame in the chamber by time observed (1 s). The average of all the speed values calculated every second throughout one deployment was considered as the average speed of one glass eel.

Ethics. At the time that this study was conducted, no permits were required by the Norwegian authorities because no fish were harmed while performing the experiments, and after use, they were either returned to the wild or sacrificed humanely. Eels were collected under research catch permit #11/11448 issued by the Norwegian Fiskeridirektoratet.

Data accessibility. All single data points are displayed in figure 4 and all the data are listed in electronic supplementary material, table S1.

Authors' contributions. A.C. designed the study; collected, analysed and interpreted the data; and wrote the paper. C.M.F.D. designed the study; collected, analysed and interpreted the data; and wrote the paper. C.B.P. designed the study, analysed and interpreted the data, wrote the paper and funded the research. C.T. collected and analysed the data. S.S. collected and analysed the data. A.B.S. designed the study, collected and interpreted the data, wrote the paper and funded the research. H.I.B. designed the study, collected and interpreted the data, wrote the paper and funded the research.

Competing interests. The authors declare that they have no competing interests.

Funding. This research was supported by the Paris Lab at the Rosenstiel School of Marine and Atmospheric Science of the University of Miami, and by funds awarded to H.I.B. by the Norwegian Institute of Marine Research project 'Fine-scale interactions in the plankton' (project no. 81529) and the Research Council of Norway (project no. 234338). A.C. was supported by the U.S. National Science Foundation NSF-OCE 1459156 to C.B.P. and by the Norwegian Institute of Marine Research (project no. 81529 to H.I.B).

Acknowledgements. We thank Josefina M. Olascoaga and Joseph E. Serafy for their valuable help with the data analysis and for providing insightful critical comments on the manuscript.

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
