## [Reviewer comments · Royal Society Open Science]

Review History

RSOS-190812.R0 (Original submission)

Review form: Reviewer 1

Is the manuscript scientifically sound in its present form?

No

Are the interpretations and conclusions justified by the results?

No

Is the language acceptable?

Yes

Is it clear how to access all supporting data?

No

Do you have any ethical concerns with this paper?

No

Have you any concerns about statistical analyses in this paper?

No

Recommendation?

Major revision is needed (please make suggestions in comments)

Comments to the Author(s)

L.299: Two collection sites (Vasseide and Stolmen) should be shown in Fig. 2.

The glass eels were collected at low tide under rocks and sediment. Were they collected during daytime? I suppose that the glass eels are nocturnal dwelling under rocks and sediment during daytime. However, the experiment in the chamber (observation by GOPRO) to observe glass eel behavior was carried out during daytime. Isn't it odd? Or, does GOPRO work fine during nighttime?

The moon moves in the same way as the sun in the new moon period. So, the glass eels may orient themselves with the sun.

How about the American eels? In order to approach north American continent, glass eels of the American eels might have to move in the opposite direction to those of European glass eels.

The authors hypothesize detection mechanism based on lunar disturbances in electrical fields.

The authors can speculate like this way, but anything could be possible.

Review form: Reviewer 2

Is the manuscript scientifically sound in its present form?

Yes

Are the interpretations and conclusions justified by the results?

Yes

Is the language acceptable?

Yes

Is it clear how to access all supporting data?

Yes

Do you have any ethical concerns with this paper?

No

Have you any concerns about statistical analyses in this paper?

No

Recommendation?

Accept with minor revision (please list in comments)

Comments to the Author(s)

The relationship between the moon cycle and the orientation of glass eels at sea

Alessandro et al.

General comments

What a pleasure it was to read such a well designed and written study on glass eel navigation at sea. My congratulations on the work – it is a fine piece of science. My comments are relatively minor I find that the manuscript is close to being publishable. Well done.

The study used a novel swimming enclosure (arena), deployed at sea off the Norwegian coast, to monitor eel orientation during daylight. Up to 57 glass eels were individually GoPro monitored within the enclosure for each of the four moon phases tested, augmented by existing data. From the results, there was glass eel orientation toward the moon azimuth at new moon.

The authors then discuss these findings in relation to other ecological events linked to moon phase (e.g. coral reef spawning) but also indicate that chemical and tidal cues may also play a role in glass eel orientation/navigation.

Several questions remain unanswered. I highlight that it is important for this manuscript to clearly identify those as: (i) glass eel navigation at night, and (ii) the true nature of the mechanism of orientation. I am satisfied that the authors do not over-extend their data here. Though there is a considerable amount of text which theoretically discusses/speculates as to the mechanism of orientation, including electric fields and Earth/Sun/Moon interactions. More specific comments about this below.

Specific comments

Line 52: The examples within the parentheses (e.g. chemical gradients, current...) seems like an unfinished thought. Please revise.

Line 142: Awkward expression – revise.

Line 174: "However, although" is a little clumsy. Revise.

Line 184: The discussion of the hypothetical mechanism for detection of the moon azimuth covers many detailed aspects, moon related electrical disturbance, the Earth's magnetotail, solar wind, spiders ballooning etc. This discussion is highly interesting and well-written but if the authors/editor's wish to reduce the manuscript length then I would encourage you to re-evaluate the utility of the different themes in this section.

Line 214: Provide a reference for electrical phenomena influencing animal behaviour.

Line 236: Your future work to monitor glass eel orientation during full moon nights using an infrared camera is an important direction and fills a key knowledge gap concerning eel movement at night. To inform your readers, I thought perhaps a one-line summary could also have appeared in your abstract.

Line 255 & 261: be consistent with capitalisation "southward" or "Southward" throughout.

Line 315: Typo "lead".

Decision letter (RSOS-190812.R0)

03-Jul-2019

Dear Mr Cresci,

The editors assigned to your paper ("The relationship between the moon cycle and the orientation of glass eels (*Anguilla anguilla*) at sea") have now received comments from reviewers. We would like you to revise your paper in accordance with the referee and Associate Editor suggestions which can be found below (not including confidential reports to the Editor). Please note this decision does not guarantee eventual acceptance.

Please submit a copy of your revised paper before 26-Jul-2019. Please note that the revision deadline will expire at 00.00am on this date. If we do not hear from you within this time then it will be assumed that the paper has been withdrawn. In exceptional circumstances, extensions

may be possible if agreed with the Editorial Office in advance. We do not allow multiple rounds of revision so we urge you to make every effort to fully address all of the comments at this stage. If deemed necessary by the Editors, your manuscript will be sent back to one or more of the original reviewers for assessment. If the original reviewers are not available, we may invite new reviewers.

- Data accessibility

If you wish to submit your supporting data or code to Dryad (<http://datadryad.org/>), or modify your current submission to dryad, please use the following link:
<http://datadryad.org/submit?journalID=RSOS&manu=RSOS-190812>

- Competing interests

- Authors' contributions

- Acknowledgements

- Funding statement

Kind regards,

Alice Power

Editorial Coordinator

on behalf of Dr Michael Tobler (Associate Editor) and Kevin Padian (Subject Editor)

Associate Editor's comments (Dr Michael Tobler):

We have received feedback from two reviewers, both of which had a positive impression of the manuscript. Both reviewers have suggestions that should help to improve the manuscript. Pending satisfactory revision, this manuscript is suitable for publication in RSOS.

Comments to Author:

Reviewers' Comments to Author:

Reviewer: 1

Comments to the Author(s)

L.299: Two collection sites (Vasseide and Stolmen) should be shown in Fig. 2.

The glass eels were collected at low tide under rocks and sediment. Were they collected during daytime? I suppose that the glass eels are nocturnal dwelling under rocks and sediment during daytime. However, the experiment in the chamber (observation by GOPRO) to observe glass eel behavior was carried out during daytime. Isn't it odd? Or, does GOPRO work fine during nighttime?

The moon moves in the same way as the sun in the new moon period. So, the glass eels may orient themselves with the sun.

How about the American eels? In order to approach north American continent, glass eels of the American eels might have to move in the opposite direction to those of European glass eels. The authors hypothesize detection mechanism based on lunar disturbances in electrical fields. The authors can speculate like this way, but anything could be possible.

Reviewer: 2

Comments to the Author(s)

The relationship between the moon cycle and the orientation of glass eels at sea

Alessandro et al.

General comments

What a pleasure it was to read such a well designed and written study on glass eel navigation at sea. My congratulations on the work – it is a fine piece of science. My comments are relatively minor I find that the manuscript is close to being publishable. Well done.

The study used a novel swimming enclosure (arena), deployed at sea off the Norwegian coast, to monitor eel orientation during daylight. Up to 57 glass eels were individually GoPro monitored within the enclosure for each of the four moon phases tested, augmented by existing data. From the results, there was glass eel orientation toward the moon azimuth at new moon.

The authors then discuss these findings in relation to other ecological events linked to moon phase (e.g. coral reef spawning) but also indicate that chemical and tidal cues may also play a role in glass eel orientation/navigation.

Several questions remain unanswered. I highlight that it is important for this manuscript to clearly identify those as: (i) glass eel navigation at night, and (ii) the true nature of the mechanism of orientation. I am satisfied that the authors do not over-extend their data here. Though there is a considerable amount of text which theoretically discusses/speculates as to the mechanism of orientation, including electric fields and Earth/Sun/Moon interactions. More specific comments about this below.

Specific comments

Line 52: The examples within the parentheses (e.g. chemical gradients, current...) seems like an unfinished thought. Please revise.

Line 142: Awkward expression – revise.

Line 174: “However, although” is a little clumsy. Revise.

Line 184: The discussion of the hypothetical mechanism for detection of the moon azimuth covers many detailed aspects, moon related electrical disturbance, the Earth’s magnetotail, solar wind, spiders ballooning etc. This discussion is highly interesting and well-written but if the authors/editor’s wish to reduce the manuscript length then I would encourage you to re-evaluate the utility of the different themes in this section.

Line 214: Provide a reference for electrical phenomena influencing animal behaviour.

Line 236: Your future work to monitor glass eel orientation during full moon nights using an infrared camera is an important direction and fills a key knowledge gap concerning eel movement at night. To inform your readers, I though perhaps a one-line summary could also have appeared in your abstract.

Line 255 & 261: be consistent with capitalisation “southward” or “Southward” throughout.

Line 315: Typo “lead”.

Author's Response to Decision Letter for (RSOS-190812.R0)

See Appendix A.

RSOS-190812.R1 (Revision)

Review form: Reviewer 2

Is the manuscript scientifically sound in its present form?

Yes

Are the interpretations and conclusions justified by the results?

Yes

Is the language acceptable?

Yes

Do you have any ethical concerns with this paper?

No

Have you any concerns about statistical analyses in this paper?

No

Recommendation?

Accept as is

Comments to the Author(s)

The authors have made a satisfactory effort to address my previous comments and thus increase the quality of the manuscript. The manuscript is now a worthwhile addition to the published literature. Well done.

Review form: Reviewer 3

Is the manuscript scientifically sound in its present form?

No

Are the interpretations and conclusions justified by the results?

No

Is the language acceptable?

Yes

Do you have any ethical concerns with this paper?

No

Have you any concerns about statistical analyses in this paper?

No

Recommendation?

Major revision is needed (please make suggestions in comments)

Comments to the Author(s)

This manuscript was previously reviewed by 2 reviewers, who did not find any major problems, but its not clear if it was checked very carefully, if any of them were eel biologists based on their comments, or I am just overly critical or am misunderstanding some things. I also find the work useful to evaluate this subject, but while starting with an open mind and being enthusiastic about a possible new discovery, I finished by being not particularly convinced of much other than the glass eels tested at new moon ebb tide preferred the south direction, which corresponded to both the direction of the moon and the sun during daytime. The Results section seems to gloss over some differences between the directions of the overall eels during the other conditions that were found to be significant (but included in the Discussion), and some environmental factors such as the current flow during new moon ebb tide or time of day of testing seem to be missing. I suppose I am worried that there could be some other explanation for this result, which is not compared to the results found about geomagnetic orientation by these same eels, and the main hypothesis is not critically evaluated very much by the authors, except to propose one possible explanation for the one clear southerly orientation. My opinion based on eel biology is that trying overly hard to discuss some kind of overall southward orientation across the species range based on these data from a far northern area is not justified, but it is for the study location. So my suggestion, since testing this hypothesis is indeed quite useful, and a nice study has been conducted, is to consider my comments and possibly make a more simple and balanced evaluation of the hypothesis without forcing the reader to believe that something important was found. Various wording changes and eliminating the big emphasis on southward swimming can solve a lot of this, but there is also a lot of work needed to standardize the formats and fonts of the figures, particularly in the supplementary file that presumably will not be edited by the journal.

Overall comments

The swimming to the south hypothesis seems like it should not be extrapolated to the whole species level in general as mentioned above, since it makes little or no sense to me at least, when considering the recruitment ecology of the European eel. That vector (south) is also just a mean direction of the moon moving across the sky, so it seems meaningless, unless I am missing something. i.e. are they following the moon position all day for a mean south direction, or they evolved a way to swim in the mean direction of the moon, which is south? Swimming south could be adaptive at times for glass eels in the far north, but the only time your glass eels clearly mostly went directly south was during new moon ebb tides and they showed little or no direct evidence of following the moon at other times despite 2 other statistical correlations. So I would suggest to focus on if they were actually swimming south at new moon ebb tides because of the moon position (and not the sun), and if there is some other possible explanation for that result that was not tested/considered. What is the current direction at the test site at those times for example. Or when were those tests done compared to the others? Seems like you might not have fully explored other possibilities, at least based on what is presently in the manuscript. The south direction during new moon looks more clear in Fig. 5 because half the eels during flood tide were added to most of the eels going south during ebb tide. That does not make the trend stronger it seems.

Another problem already mentioned is that this manuscript does not compare to their previous paper that showed that glass eel orientation was affected by shifts in the geomagnetic field. So how do the two hypotheses interact? Only mentioned in passing it seems, and I don't have time to check it right now.

Maybe I am not thinking clearly, but Fig. S6 seems to show the same statistically significant pattern of orientation occurred in the direction of the sun during new moon. Less so for first

quarter. So why is the possibility of orienting to the sun during the tests during new moon not evaluated?

Figure S7 shows the light intensity is not very similar among the 4 lunar phase periods, with first quarter being highest, new moon next, but with the greatest variability in light levels. So I suppose, the question is, how can the direction of the sun be excluded. Was time of day controlled or tested? Sorry if I missed that, or cloud cover vs sunny? Ok, at least the next figure compares light levels.

Comments by line number

L44 flows north on which tide? Flood or ebb?

L57 The spawning is synchronized with new moon, the hatching is just a result of when spawning occurred, so different wording is needed.

Table 1. Please clarify in the table heading what "moon illumination" refers to? i.e. the glass eels were tested during the day, and the moon can be in the sky sometimes during the day. So this means what? Percent of time the moon is out at night or during day. Readers should not need to read the methods to understand major things.

L105 It seems like you should define what you mean by azimuth even though that is a standard term. Without clearly knowing what that is, this study is meaningless.

L107 regarding "common orientation, which was southward" and other subsequent statements, much of this seems over simplified and confusing. 4C Flood middle shows SE and SW azimuth positions, not directly "south", and 4C left shows 20 eels oriented in the northern half of the directions, and 17 oriented in the southerly directions, with more in the SE with most of the azimuths (that is not shown as significant). But that becomes significant when you compare using the differences between orientation difference between orientation direction and azimuth? Apparently, but that looks not far from random chance to the skeptical observer, so perhaps you should not be so casual in how you describe these results. i.e. direction is not significant, but the difference is, might be mentioned at least.

4C ebb is convincing for orientation direction matching the azimuth direction, but that is the only clear match I see, compared to the very short and optimistic text.

This section needs revision, since for example "During flood/new moon (Fig. 4C), tests were performed while the moon azimuths were in opposite directions (East-West) (Fig 4C. 110 column ii)." Only refers to the flood case, not the new moon case. But the sentence starts mentioning both.

L111 There was no common orientation to south?

L113 regarding "The orientation towards the new moon azimuth", I see one clear example of that for new moon. So a statement more like "The apparent orientation in similar directions as the new moon azimuth found statistically in both flood and ebb....."

L115 "towards the moon"? you were comparing to the azimuth including above and below the horizon, so that's a little bit sloppy wording maybe. Toward the direction of xxx.

L118 "when the moon starts becoming brighter", again that is at night, you tested during the day, so please clarify what you are referring to.

L120 4D Flood, the glass eels went to the SE and the azimuths are to the NE, so probably you should mention that and not just point to the statistical result? Seems like a low bar for significance somehow.

L121 I don't see what "both common orientations" refers to. 4D flood and ebb eel orientations are not at all similar and only one is to the south.

I was successfully confused by trying to compare the text to the figures, so the text should describe what the reader can see in the figure better, carefully referring to if flood or ebb etc. is being referred to, since much of the text seems unclear, and statistics is not always the only thing that matters in terms of if the reader is going to accept what the authors are saying or not.

Discussion

L142 What I see in Fig. 4, and a more accurate start to the Discussion could be something like this: Glass eels oriented in the same direction as the azimuth of the moon at sea only during specific phases of the lunar cycle, and some of the statistical correlations were only found when the orientation directions of individual glass eels were compared to the directions of the moon azimuth. The clearest correspondence occurred during new moon ebb tides when the glass eels southerly orientation was in the same direction as the moon above the horizon. However, in the other cases of significant correlations, the overall glass eel orientation direction, did not clearly match the direction of the azimuth of the moon. When the individual eel orientations were analyzed.... Significant correlations were found for the xxx, xxx, xxx cases...

The point is, this first paragraph of the Discussion gives very little information about the actual positive results of the study, but seems to be trying to provide some kind of general summary with only one actual statement about the statistically significant results outcome, which is "a significant orientation was observed when the moon was above the horizon at new moon". i.e. what about first quarter, that was significant also according the results during flood tide? How clear was each of the 3 positive results? There are a few statements later that clarify some things, but this seems like an incomplete start to the discussion after a confusing Results section, before going on to get into the literature.

L158 Regarding, "The orientation of glass eels in our study was related to the moon's azimuth and it's position relative to the horizon", yes it was statistically related in 3 out of 8 of the conditions, and only clearly in one case, so its hard to see how such a simple statement can be used, without more detailed wording.

L183 Yes, so more detailed descriptions like this should also be in the Results and then mentioned generally like this in the Discussion.

L187 New moon flood seems no more clear than first quarter flood, so why only mention new moon. i.e. the statistical result either means something or it does not. The clear result is new moon ebb obviously.

I don't see the logic of saying "only when the moon was invisible to them" as support for the mechanism cant be visual. The tests were in daytime, so the moon was always invisible to them, and they are under water, so they will never see the moon regardless of drifting at day or night. At night they can see "moonlight" but not the moon itself in most cases. Please clarify if what you are talking about, such as during your testing, or in general unrelated to the testing periods.

L206 clarify the increase is compared to what?

L232 need to add the reference number again and not just the author names.

L243 A single sentence paragraph is appropriate? ...could be play a role... is a typo

L263 I am losing the logic here since you found no trend during full moon. And the concept of swimming south also needs clarifying, since the text is saying that is a mean direction of the moon across the whole time the moon is up. So the eels start out swimming east, then SE, S, SW and then almost west following the moon direction, resulting in a mean south direction? That seems strange and maladapted. This raises the question then about why you don't discuss the time of day of your tests? That would affect where the moon (and sun) is in the sky.

L267 Recruitment at new and full moon is also thought to be related to strong tidal flows isn't it?

-species of glass eels- not eel

Figures

It seems like the figures were not checked much for basic consistency. Maybe it's assumed to be the copy editors job for the main manuscript, but for the supplementary file it seems the authors should try to set a better example for young scientists, unless the journal will edit that also. I will mention a few examples or general things to check if you care about this.

Figure 2: Fonts range from readable to microscopic. Including for the same y-axis label of "Latitude".

The size range issue applies to almost all figures, but there is also obvious variability of use of bold text, types of fonts, use of capital first letters in most of the figures.

Figure 5 has a technical issue of the flows of the Gulf Stream being incorrect, or bad even for a schematic drawing. The most obvious is current flowing over what appears to be the continental shelf off Atlantic Canada. However, unless I am really missing something here, I would suggest that this figure is not justified to include, and crosses beyond what your data show.

A few examples of what to check for in all Supplementary figures:

Figure S2 gives the units, but not what it is.

Figure S3 has much larger font than S2

Figure S4 has bold x-axis labels, the y-axis does not

Figure S8 lower case y-axis label, upper case x-axis

Decision letter (RSOS-190812.R1)

23-Sep-2019

Dear Mr Cresci:

On behalf of the Editors, I am pleased to inform you that your Manuscript RSOS-190812.R1 entitled "The relationship between the moon cycle and the orientation of glass eels (*Anguilla anguilla*) at sea" has been accepted for publication in Royal Society Open Science subject to minor revision in accordance with the referee suggestions. Please find the referees' comments at the end of this email.

The reviewers and Subject Editor have recommended publication, but also suggest some minor

revisions to your manuscript. Therefore, I invite you to respond to the comments and revise your manuscript.

- Ethics statement

- Data accessibility

If you wish to submit your supporting data or code to Dryad (<http://datadryad.org/>), or modify your current submission to dryad, please use the following link:
<http://datadryad.org/submit?journalID=RSOS&manu=RSOS-190812.R1>

- Competing interests

- Authors' contributions

- Acknowledgements

- Funding statement

Please note that we cannot publish your manuscript without these end statements included. We

have included a screenshot example of the end statements for reference. If you feel that a given heading is not relevant to your paper, please nevertheless include the heading and explicitly state that it is not relevant to your work.

Because the schedule for publication is very tight, it is a condition of publication that you submit the revised version of your manuscript before 02-Oct-2019. Please note that the revision deadline will expire at 00.00am on this date. If you do not think you will be able to meet this date please let me know immediately.

Kind regards,
Andrew Dunn
Royal Society Open Science Editorial Office
Royal Society Open Science

on behalf of Dr Michael Tobler (Associate Editor) and Kevin Padian (Subject Editor)
openscience@royalsociety.org

Associate Editor Comments to Author (Dr Michael Tobler):

Associate Editor: 1

Comments to the Author:

We have received the feedback from two reviewers that were overall very positive about the revised manuscript. One reviewer has provided detailed additional feedback that should be relatively easy to address and will improve the readability of the paper. Upon revisions, this paper should fit the requirements for publication in RSOS.

Reviewer comments to Author:

Reviewer: 2

Comments to the Author(s)

The authors have made a satisfactory effort to address my previous comments and thus increase the quality of the manuscript. The manuscript is now a worthwhile addition to the published literature. Well done.

Reviewer: 3

Comments to the Author(s)

This manuscript was previously reviewed by 2 reviewers, who did not find any major problems, but it's not clear if it was checked very carefully, if any of them were eel biologists based on their comments, or I am just overly critical or am misunderstanding some things. I also find the work useful to evaluate this subject, but while starting with an open mind and being enthusiastic about a possible new discovery, I finished by being not particularly convinced of much other than the glass eels tested at new moon ebb tide preferred the south direction, which corresponded to both the direction of the moon and the sun during daytime. The Results section seems to gloss over some differences between the directions of the overall eels during the other conditions that were found to be significant (but included in the Discussion), and some environmental factors such as the current flow during new moon ebb tide or time of day of testing seem to be missing. I suppose I am worried that there could be some other explanation for this result, which is not compared to the results found about geomagnetic orientation by these same eels, and the main hypothesis is not critically evaluated very much by the authors, except to propose one possible explanation for the one clear southerly orientation. My opinion based on eel biology is that trying overly hard to discuss some kind of overall southward orientation across the species range based on these data from a far northern area is not justified, but it is for the study location. So my suggestion, since testing this hypothesis is indeed quite useful, and a nice study has been conducted, is to consider my comments and possibly make a more simple and balanced evaluation of the hypothesis without forcing the reader to believe that something important was found. Various wording changes and eliminating the big emphasis on southward swimming can solve a lot of this, but there is also a lot of work needed to standardize the formats and fonts of the figures, particularly in the supplementary file that presumably will not be edited by the journal.

Overall comments

The swimming to the south hypothesis seems like it should not be extrapolated to the whole species level in general as mentioned above, since it makes little or no sense to me at least, when considering the recruitment ecology of the European eel. That vector (south) is also just a mean direction of the moon moving across the sky, so it seems meaningless, unless I am missing something. i.e. are they following the moon position all day for a mean south direction, or they evolved a way to swim in the mean direction of the moon, which is south? Swimming south could be adaptive at times for glass eels in the far north, but the only time your glass eels clearly mostly went directly south was during new moon ebb tides and they showed little or no direct evidence of following the moon at other times despite 2 other statistical correlations. So I would suggest to focus on if they were actually swimming south at new moon ebb tides because of the moon position (and not the sun), and if there is some other possible explanation for that result that was not tested/considered. What is the current direction at the test site at those times for example. Or when were those tests done compared to the others? Seems like you might not have fully explored other possibilities, at least based on what is presently in the manuscript. The south direction during new moon looks more clear in Fig. 5 because half the eels during flood tide were added to most of the eels going south during ebb tide. That does not make the trend stronger it seems.

Another problem already mentioned is that this manuscript does not compare to their previous paper that showed that glass eel orientation was affected by shifts in the geomagnetic field. So how do the two hypotheses interact? Only mentioned in passing it seems, and I don't have time to check it right now.

Maybe I am not thinking clearly, but Fig. S6 seems to show the same statistically significant pattern of orientation occurred in the direction of the sun during new moon. Less so for first quarter. So why is the possibility of orienting to the sun during the tests during new moon not evaluated?

Figure S7 shows the light intensity is not very similar among the 4 lunar phase periods, with first quarter being highest, new moon next, but with the greatest variability in light levels. So I suppose, the question is, how can the direction of the sun be excluded. Was time of day controlled or tested? Sorry if I missed that, or cloud cover vs sunny? Ok, at least the next figure compares light levels.

Comments by line number

L44 flows north on which tide? Flood or ebb?

L57 The spawning is synchronized with new moon, the hatching is just a result of when spawning occurred, so different wording is needed.

Table 1. Please clarify in the table heading what "moon illumination" refers to? i.e. the glass eels were tested during the day, and the moon can be in the sky sometimes during the day. So this means what? Percent of time the moon is out at night or during day. Readers should not need to read the methods to understand major things.

L105 It seems like you should define what you mean by azimuth even though that is a standard term. Without clearly knowing what that is, this study is meaningless.

L107 regarding "common orientation, which was southward" and other subsequent statements, much of this seems over simplified and confusing. 4C Flood middle shows SE and SW azimuth positions, not directly "south", and 4C left shows 20 eels oriented in the northern half of the directions, and 17 oriented in the southerly directions, with more in the SE with most of the

azimuths (that is not shown as significant). But that becomes significant when you compare using the differences between orientation difference between orientation direction and azimuth? Apparently, but that looks not far from random chance to the skeptical observer, so perhaps you should not be so casual in how you describe these results. i.e. direction is not significant, but the difference is, might be mentioned at least.

4C ebb is convincing for orientation direction matching the azimuth direction, but that is the only clear match I see, compared to the very short and optimistic text.

This section needs revision, since for example "During flood/new moon (Fig. 4C), tests were performed while the moon azimuths were in opposite directions (East-West) (Fig 4C. 110 column ii)." Only refers to the flood case, not the new moon case. But the sentence starts mentioning both.

L111 There was no common orientation to south?

L113 regarding "The orientation towards the new moon azimuth", I see one clear example of that for new moon. So a statement more like "The apparent orientation in similar directions as the new moon azimuth found statistically in both flood and ebb....."

L115 "towards the moon"? you were comparing to the azimuth including above and below the horizon, so that's a little bit sloppy wording maybe. Toward the direction of xxx.

L118 "when the moon starts becoming brighter", again that is at night, you tested during the day, so please clarify what you are referring to.

L120 4D Flood, the glass eels went to the SE and the azimuths are to the NE, so probably you should mention that and not just point to the statistical result? Seems like a low bar for significance somehow.

L121 I don't see what "both common orientations" refers to. 4D flood and ebb eel orientations are not at all similar and only one is to the south.

I was successfully confused by trying to compare the text to the figures, so the text should describe what the reader can see in the figure better, carefully referring to if flood or ebb etc. is being referred to, since much of the text seems unclear, and statistics is not always the only thing that matters in terms of if the reader is going to accept what the authors are saying or not.

Discussion

L142 What I see in Fig. 4, and a more accurate start to the Discussion could be something like this: Glass eels oriented in the same direction as the azimuth of the moon at sea only during specific phases of the lunar cycle, and some of the statistical correlations were only found when the orientation directions of individual glass eels were compared to the directions of the moon azimuth. The clearest correspondence occurred during new moon ebb tides when the glass eels southerly orientation was in the same direction as the moon above the horizon. However, in the other cases of significant correlations, the overall glass eel orientation direction, did not clearly match the direction of the azimuth of the moon. When the individual eel orientations were analyzed.... Significant correlations were found for the xxx, xxx, xxx cases...

The point is, this first paragraph of the Discussion gives very little information about the actual positive results of the study, but seems to be trying to provide some kind of general summary with only one actual statement about the statistically significant results outcome, which is "a significant orientation was observed when the moon was above the horizon at new moon". i.e.

what about first quarter, that was significant also according the results during flood tide? How clear was each of the 3 positive results? There are a few statements later that clarify some things, but this seems like an incomplete start to the discussion after a confusing Results section, before going on to get into the literature.

L158 Regarding, "The orientation of glass eels in our study was related to the moon's azimuth and it's position relative to the horizon", yes it was statistically related in 3 out of 8 of the conditions, and only clearly in one case, so its hard to see how such a simple statement can be used, without more detailed wording.

L183 Yes, so more detailed descriptions like this should also be in the Results and then mentioned generally like this in the Discussion.

L187 New moon flood seems no more clear than first quarter flood, so why only mention new moon. i.e. the statistical result either means something or it does not. The clear result is new moon ebb obviously.

I don't see the logic of saying "only when the moon was invisible to them" as support for the mechanism cant be visual. The tests were in daytime, so the moon was always invisible to them, and they are under water, so they will never see the moon regardless of drifting at day or night. At night they can see "moonlight" but not the moon itself in most cases. Please clarify if what you are talking about, such as during your testing, or in general unrelated to the testing periods.

L206 clarify the increase is compared to what?

L232 need to add the reference number again and not just the author names.

L243 A single sentence paragraph is appropriate? ...could be play a role... is a typo

L263 I am losing the logic here since you found no trend during full moon. And the concept of swimming south also needs clarifying, since the text is saying that is a mean direction of the moon across the whole time the moon is up. So the eels start out swimming east, then SE, S, SW and then almost west following the moon direction, resulting in a mean south direction? That seems strange and maladapted. This raises the question then about why you don't discuss the time of day of your tests? That would affect where the moon (and sun) is in the sky.

L267 Recruitment at new and full moon is also thought to be related to strong tidal flows isn't it?

-species of glass eels- not eel

Figures

It seems like the figures were not checked much for basic consistency. Maybe it's assumed to be the copy editors job for the main manuscript, but for the supplementary file it seems the authors should try to set a better example for young scientists, unless the journal will edit that also. I will mention a few examples or general things to check if you care about this.

Figure 2: Fonts range from readable to microscopic. Including for the same y-axis label of "Latitude".

The size range issue applies to amost all figures, but there is also obvious variability of use of bold text, types of fonts, use of capital first letters in most of the figures.

Figure 5 has a technical issue of the flows of the Gulf Stream being incorrect, or bad even for a

schematic drawing. The most obvious is current flowing over what appears to be the continental shelf off Atlantic Canada. However, unless I am really missing something here, I would suggest that this figure is not justified to include, and crosses beyond what your data show.

A few examples of what to check for in all Supplementary figures:

Figure S2 gives the units, but not what it is.

Figure S3 has much larger font than S2

Figure S4 has bold x-axis labels, the y-axis does not

Figure S8 lower case y-axis label, upper case x-axis

Author's Response to Decision Letter for (RSOS-190812.R1)

See Appendix B.

Decision letter (RSOS-190812.R2)

01-Oct-2019

Dear Mr Cresci,

I am pleased to inform you that your manuscript entitled "The relationship between the moon cycle and the orientation of glass eels (*Anguilla anguilla*) at sea" is now accepted for publication in Royal Society Open Science.

on behalf of Dr Michael Tobler (Associate Editor) and Kevin Padian (Subject Editor)
openscience@royalsociety.org

Appendix A

REPLIES TO REVIEWERS

We thank the reviewers for the helpful comments. We addressed all the comments and edited the manuscript accordingly. We are glad that both reviewers found the work valuable.

We reply to each comment below. Our replies are in blue.

Reviewer: 1

Comments to the Author(s)

L.299: Two collection sites (Vasseide and Stolmen) should be shown in Fig. 2.

Answer: We added the collection sites to Figure 2 and we added the reference to the figure in the text (Line 301).

The glass eels were collected at low tide under rocks and sediment. Were they collected during daytime? I suppose that the glass eels are nocturnal dwelling under rocks and sediment during daytime.

Answer: Glass eels were collected at low tide during daytime. These were the best conditions to catch glass eels as the water was shallow and the sunlight allowed us to see the them. Glass eels' behavior varies both according to light and tide. When they are at the estuary they try to avoid being carried offshore by the current during the ebb/low tide, and that was why they hid in the sediment or under rocks during that tidal phase.

However, the experiment in the chamber (observation by GOPRO) to observe glass eel behavior was carried out during daytime. Isn't it odd? Or, does GOPRO work fine during nighttime?

Answer: during their marine phase, glass eels migrate both during the day and during the night, swimming and transported by the currents. Our hypothesis was that their migratory behavior could be influenced by the lunar cycle independently from the light conditions. This is why we tested their orientation during the day (and we could indeed see such effect). However, we will test their orientation during the night in future work as we stated in the discussion. GOPROs work during the night but they need the IR filters to be removed. Furthermore, the DISC chambers would need infrared light systems to perform such tests (with additional external batteries too). We intend to use this setup for future studies on behavior at night.

The moon moves in the same way as the sun in the new moon period. So, the glass eels may orient themselves with the sun.

Answer: we ruled out the potential role of the sun analyzing glass eels' orientation with respect to the sun azimuth and comparing light intensity levels (using HOBO sensors *in situ*) between the different moon phases. We described such analysis in the Supplementary Information Figures S5-S8, which clearly shows that the sun-related orientation is an artifact of the lunar-driven orientation.

How about the American eels? In order to approach north American continent, glass eels of the American eels might have to move in the opposite direction to those of European glass eels.

Answer: Because we performed these experiments on the European eel, we did not want to speculate on other species. However, this is an interesting question. If American eels had the same lunar-related orientation behavior, they would also swim south on average during new moon. As the Gulf Stream flows N-NE, this behavior could potentially help their migration.

The authors hypothesize detection mechanism based on lunar disturbances in electrical fields. The authors can speculate like this way, but anything could be possible.

Answer: We propose this detection mechanism as a hypothesis (thus it is admittedly speculative) based on the observed results on the behavior of glass eels. We believe that due to glass eels' electro sensitivity and magnetic sense, the global-scale electrical disturbances of the moon could be the cue that glass eels use to orient towards the moon. The best orientation response was observed during new moon, during daytime, when the moon was invisible. Thus, the mechanism involved cannot be visual.

Reviewer: 2

Comments to the Author(s)

The relationship between the moon cycle and the orientation of glass eels at sea

Alessandro et al.

General comments

What a pleasure it was to read such a well designed and written study on glass eel navigation at sea. My congratulations on the work – it is a fine piece of science. My comments are relatively minor I find that the manuscript is close to being publishable. Well done.

The study used a novel swimming enclosure (arena), deployed at sea off the Norwegian coast, to monitor eel orientation during daylight. Up to 57 glass eels were individually GoPro monitored within the enclosure for each of the four moon phases tested, augmented by existing data. From the results, there was glass eel orientation toward the moon azimuth at new moon.

The authors then discuss these findings in relation to other ecological events linked to moon phase (e.g. coral reef spawning) but also indicate that chemical and tidal cues may also play a role in glass eel orientation/navigation.

Several questions remain unanswered. I highlight that it is important for this manuscript to clearly identify those as: (i) glass eel navigation at night, and (ii) the true nature of the mechanism of orientation. I am satisfied that the authors do not over-extend their data here. Though there is a considerable amount of text which theoretically discusses/speculates as to the mechanism of orientation, including electric fields and Earth/Sun/Moon interactions. More specific comments about this below.

Specific comments

Line 52: The examples within the parentheses (e.g. chemical gradients, current...) seems like an unfinished thought. Please revise.

Answer: Line 52: "(e.g. chemical gradients, current...)" replaced by "(e.g. salinity gradients, odor plumes and water currents)".

Line 142: Awkward expression – revise.

Answer: Line 142: We edited the sentence as follow “Specifically, a significant orientation was observed when the moon was above the horizon at new moon. With these conditions the visibility of the moon is minimal (at this time, the moon is invisible).”

Line 174: “However, although” is a little clumsy. Revise.

Answer: Line 175: sentence edited as “However, the presence of the moon above the horizon is important but it is not the only factor that plays a role”

Line 184: The discussion of the hypothetical mechanism for detection of the moon azimuth covers many detailed aspects, moon related electrical disturbance, the Earth’s magnetotail, solar wind, spiders ballooning etc. This discussion is highly interesting and well-written but if the authors/editor’s wish to reduce the manuscript length then I would encourage you to re-evaluate the utility of the different themes in this section.

Answer: We believe that this section is important for follow-up research. It could inspire future work to investigate the mechanism that glass eels use to detect the direction of the moon. We also believe that this section also highlights the interdisciplinarity of this research, which adds value to this work for a broader audience from multiple disciplines. As the journal does not have word limits, we would rather not further shorten this section, as every topic is important for the logical flow of the hypothesis.

Line 214: Provide a reference for electrical phenomena influencing animal behaviour.

Answer: Line 214: we adder the following references:

-Bevington M. 2015 Lunar biological effects and the magnetosphere

-Morley EL, Robert D. 2018 Electric Fields Elicit Ballooning in Spiders.

-Bhattacharjee C, Bradley P, Smith M, Scally AJ, Wilson BJ. 2000 Do animals bite more during a full moon? Retrospective observational analysis.

-Mercier A, Zhao Sun, Baillon S, Hamel JF. 2011 Lunar rhythms in the deep sea: Evidence from the reproductive periodicity of several marine invertebrates.

Line 236: Your future work to monitor glass eel orientation during full moon nights using an infrared camera is an important direction and fills a key knowledge gap concerning eel movement at night. To inform your readers, I though perhaps a one-line summary could also have appeared in your abstract.

Answer: we understand the point raised by the reviewer, but we believe that a summary on future work would not really fit into the abstract. If this was a review article, we would follow this advise but because this is a research article, we rather leave references to future work in the discussion. Specifically, at line 235 we state:

“In future work we intend to investigate whether glass eels orient towards the moon during full moon nights using infrared cameras, and conduct experiments on glass eels’ sensitivity to electric fields comparable to those caused by the moon.”

Line 255 & 261: be consistent with capitalisation “southward” or “Southward” throughout.

Answer: edited to “southward”.

Line 315: Typo “lead”.

Answer: “lead” replaced by “lid”.

Appendix B

We thank the reviewer for the comments which helped us improve the overall clarity of the manuscript. We revised the manuscript following the suggestions of the reviewer.

We reply to each comment below. The reviewer's comments are in black and our replies in blue.

Reviewer: 3

Overall comments

The swimming to the south hypothesis seems like it should not be extrapolated to the whole species level in general as mentioned above, since it makes little or no sense to me at least, when considering the recruitment ecology of the European eel.

The European eel constitutes a single, panmictic population. This means that all of the eels hatching in the Sargasso Sea have, in theory, equal chances to recruit to any area from northern Norway to the Mediterranean Sea. The present study was conducted to investigate how the orientation of glass eels is influenced by the moon, which is a strong and stable cue, available in the whole area of distribution of the eel, from Norway to the Mediterranean Sea. It is, therefore, justified to assume that the behavior observed in this study is shared by other glass eels in other regions, which belong to the same panmictic population. This would be different if the species studied was, for example, salmon, which has distinct populations and their orientation behavior depends upon the natal stream.

However, we accept that if this reviewer had this concern so may some readers and, therefore, we added the following statement at the end of the discussion:

Line 307 : " However, it is also possible that glass eels might display differences in lunar-related orientation behavior depending on the recruitment area. We will test these scenarios and hypotheses in future work."

That vector (south) is also just a mean direction of the moon moving across the sky, so it seems meaningless, unless I am missing something. i.e. are they following the moon position all day for a mean south direction, or they evolved a way to swim in the mean direction of the moon, which is south? Swimming south could be adaptive at times for glass eels in the far north, but the only time your glass eels clearly mostly went directly south was during new moon ebb tides and they showed little or no direct evidence of following the moon at other times despite 2 other statistical correlations.

Taken as a whole, the statistical analysis shows that the orientation direction of the glass eels is towards the South at new moon, but not during the other lunar phases. Concerning the orientation during new moon, we observed a statistically significant orientation towards the moon regardless of the tidal phase (Fig 4C. column iii). Glass eels oriented South during the ebb tide when the moon was South (Fig. 4C ii). However, during flood tide the moon azimuth was split East-West and if glass eels were swimming South independently from the moon azimuth we would expect to see orientation to the South also during flood tide, which was not the case. Thus, concerning the reviewer's question: glass eels do not orient towards the moon because they go South by default, rather they follow the moon and, when this is south, they go South. Our analysis shows that the orientation of glass eels towards the moon azimuth during new moon does not depend on the tidal phase, and that is highly significant during the entire new moon phase (Fig. 5A iii).

So I would suggest to focus on if they were actually swimming south at new moon ebb tides because of the moon position (and not the sun), and if there is some other possible explanation for that result that was not tested/considered. What is the current direction at the test site at those times for example. Or when were those tests done compared to the others? Seems like you might not have fully explored other possibilities, at least based on what is presently in the manuscript. The south direction during new moon looks more clear in Fig. 5 because half the eels during flood tide were added to most of the eels going south during ebb tide. That does not make the trend stronger it seems.

We provided a very detailed analysis to rule out the possible role of the sun in the orientation of glass eels. We show the results of the analysis in the supplementary information (Section 2 and Fig. S5 – S8). Glass eels appeared to orient towards the sun only during new moon. However, if glass eels were following the sun, we would expect to see sun-related orientation during any lunar phase as the light intensity (caused by the sun) between new moon and other lunar phases did not differ statistically. This was not the case.

With respect to currents: the glass eels were tested in a drifting chamber that was specifically designed to drift together with the current (DISC, Drifting In Situ Chamber). This means that glass eels were moving together with the current which, therefore, had no velocity relative to their body. Thus, it was not possible for them to detect it. This is explained thoroughly in the Methods. Figure 5 presents the orientation analysis of all eels tested at new moon to show that the orientation towards the moon is highly significant throughout the lunar phase ($P = 0.000006$; Fig. 5A iii).

Another problem already mentioned is that this manuscript does not compare to their previous paper that showed that glass eel orientation was affected by shifts in the geomagnetic field. So how do the two hypotheses interact? Only mentioned in passing it seems, and I don't have time to check it right now.

During the pelagic phase of the glass eel migration, the direction that they follow is guided by the moon during new moon. In our earlier article, which reported that glass eels possess a magnetic compass, we showed that glass eels use the magnetic field of the earth to orient. Our thinking is that the lunar-related orientation described here, and the magnetic orientation, are both part of a complex and multifaceted orientation mechanism. We address this in the Discussion as follows:

Line 176: "Moon-related orientation might work together with the magnetic compass during the pelagic step of the migration over the continental shelf. One possibility is that the moon might serve as a directional stimulus and the magnetic compass as a frame of reference. Another possibility is that the two systems might alternate such that when the moon is undetectable eels might use the compass to maintain a course learned during previous new moon phases. These hypotheses will be tested in future research."

Maybe I am not thinking clearly, but Fig. S6 seems to show the same statistically significant pattern of orientation occurred in the direction of the sun during new moon. Less so for first quarter. So why is the possibility of orienting to the sun during the tests during new moon not evaluated?

We describe why that analysis rules out the sun as a possible cue followed by the eels in the section 2 of the supplemental information. We measured the sunlight using sensors during the tests during each one

of the lunar phases. The sun was available for most of the lunar phases and the sunlight did not differ statistically between new moon and the other phases. If the glass eels were following the sun, they would orient towards the sun independently of the moon phase (as the sun doesn't change features between different moon phases like the moon does). However, we observed orientation only during new moon and partially during first quarter. This indicates that this apparent sun-related orientation is an artifact caused by the lunar-related orientation.

Figure S7 shows the light intensity is not very similar among the 4 lunar phase periods, with first quarter being highest, new moon next, but with the greatest variability in light levels. So I suppose, the question is, how can the direction of the sun be excluded. Was time of day controlled or tested? Sorry if I missed that, or cloud cover vs sunny? Ok, at least the next figure compares light levels.

We compare the light levels between the moon phases statistically in Fig. S8. The light intensity during new moon (when we observed orientation) does not differ compared to full moon and third quarter. Although sunlight is present during these phases, glass eels do not orient towards the sun.

Comments by line number

L44 flows north on which tide? Flood or ebb?

We added "during ebb tide"

L57 The spawning is synchronized with new moon, the hatching is just a result of when spawning occurred, so different wording is needed.

We replaced "hatching" with "spawning"

Table 1. Please clarify in the table heading what "moon illumination" refers to? i.e. the glass eels were tested during the day, and the moon can be in the sky sometimes during the day. So this means what? Percent of time the moon is out at night or during day. Readers should not need to read the methods to understand major things.

We agree "moon illumination" was confusing. We replaced it with "Percentage of moon visible".

L105 It seems like you should define what you mean by azimuth even though that is a standard term. Without clearly knowing what that is, this study is meaningless.

We now define "azimuth" in the introduction (Line 77) as

"(the angle between the North and the vertical projection of a celestial body onto the horizon)"

L107 regarding “common orientation, which was southward” and other subsequent statements, much of this seems over simplified and confusing. 4C Flood middle shows SE and SW azimuth positions, not directly “south”, and 4C left shows 20 eels oriented in the northern half of the directions, and 17 oriented in the southerly directions, with more in the SE with most of the azimuths (that is not shown as significant). But that becomes significant when you compare using the differences between orientation difference between orientation direction and azimuth? Apparently, but that looks not far from random chance to the skeptical observer, so perhaps you should not be so casual in how you describe these results. i.e. direction is not significant, but the difference is, might be mentioned at least.

We stated that there is no common orientation during new moon/flood tide. However, we accept that if this reviewer had this concern so may some readers and, therefore, we replaced “common orientation, which was southward” with “common orientation towards the moon azimuth”, which was indeed observed at new moon during both ebb and flood tides.

4C ebb is convincing for orientation direction matching the azimuth direction, but that is the only clear match I see, compared to the very short and optimistic text.

This section needs revision, since for example “During flood/new moon (Fig. 4C), tests were performed while the moon azimuths were in opposite directions (East-West) (Fig 4C. 110 column ii).” Only refers to the flood case, not the new moon case. But the sentence starts mentioning both.

We realized that a description of the results during new moon/ebb tide was missing and we thank the reviewer for pointing that out. We edited the paragraph and now make a clear distinction between the results for flood and ebb tides at new moon.

Line 108 :“During the ebb tide at new moon, glass eels oriented towards the moon azimuth (Fig 4C, column iii), the moon azimuth was south (Fig 4C, column ii), and glass eels oriented to the south (Fig 4C, column i). During the flood tide at new moon (Fig. 4C), tests were performed while the moon azimuths were in opposite directions (East-West) (Fig 4C, column ii). Under these conditions, the behavior of glass eels did not change; they still oriented towards the azimuth of the moon (Fig. 4C, column iii). However, there was no common orientation towards magnetic south (Fig. 4C, column i).”

L111 There was no common orientation to south?

See the preceding response.

L113 regarding “The orientation towards the new moon azimuth”, I see one clear example of that for new moon. So a statement more like “The apparent orientation in similar directions as the new moon azimuth found statistically in both flood and ebb.....”

We don't understand the point raised by the reviewer as we do state that the orientation is towards the

“new moon azimuth”. The results described in this paragraph are only about the new moon data, as stated at the beginning of the paragraph.

L115 “towards the moon”? you were comparing to the azimuth including above and below the horizon, so that’s a little bit sloppy wording maybe. Toward the direction of xxx.

We replaced “towards the moon” with “towards the azimuth of the moon”. However, this specific result is about the new moon data, when the moon was always above the horizon.

L118 “when the moon starts becoming brighter”, again that is at night, you tested during the day, so please clarify what you are referring to.

We replaced “when the moon starts becoming brighter” with “when the percentage of moon visible to the eye increases”

L120 4D Flood, the glass eels went to the SE and the azimuths are to the NE, so probably you should mention that and not just point to the statistical result? Seems like a low bar for significance somehow.

We now clarify “the moon azimuth was towards the Northeast and a common orientation towards the Southeast was observed”

L121 I don’t see what “both common orientations” refers to. 4D flood and ebb eel orientations are not at all similar and only one is to the south.

We were not referring to both tides, but were describing the orientation results during flood tide only. We agree with the reviewer that this was confusing, and we edited the sentence to make it clearer.

Line 124: “The common orientation (Fig. 4D, column i and iii) towards magnetic south and the moon azimuth observed during flood tide disappeared during the ebb tide, when the moon fell below the horizon (Fig. 4D, column ii)”

I was successfully confused by trying to compare the text to the figures, so the text should describe what the reader can see in the figure better, carefully referring to if flood or ebb etc. is being referred to, since much of the text seems unclear, and statistics is not always the only thing that matters in terms of if the reader is going to accept what the authors are saying or not.

We edited the results section following all of the suggestions of the reviewer (see all the comments above). We trust that this is now clearer.

Discussion

L142 What I see in Fig. 4, and a more accurate start to the Discussion could be something like this: Glass eels oriented in the same direction as the azimuth of the moon at sea only during specific phases of the lunar cycle, and some of the statistical correlations were only found when the orientation directions of individual glass eels were compared to the directions of the moon azimuth. The clearest correspondence occurred during new moon ebb tides when the glass eels southerly orientation was in the same direction as the moon above the horizon. However, in the other cases of significant correlations, the overall glass eel orientation direction, did not clearly match the direction of the azimuth of the moon. When the individual eel orientations were analyzed.... Significant correlations were found for the xxx, xxx, xxx cases...

The point is, this first paragraph of the Discussion gives very little information about the actual positive results of the study, but seems to be trying to provide some kind of general summary with only one actual statement about the statistically significant results outcome, which is “a significant orientation was observed when the moon was above the horizon at new moon”. i.e. what about first quarter, that was significant also according the results during flood tide? How clear was each of the 3 positive results? There are a few statements later that clarify some things, but this seems like an incomplete start to the discussion after a confusing Results section, before going on to get into the literature.

While we already describe these results later in the discussion, we agree with the reviewer that a more detailed summary of the results was missing at the beginning of the discussion. Therefore, we have added the following lines to the first paragraph of the discussion.

Line 148: “The clearest response was observed during new moon/ebb tide, when glass eels also oriented to the South, but it was different during new moon/flood tide, when the azimuth of the moon was split between east and west. Under these conditions, glass eels oriented towards the moon azimuth but did not orient with respect to the North. Significant patterns in orientation direction were also observed during first quarter, but only during flood tide, when the moon was above the line of the horizon. However, the orientation response was 75° from the moon azimuth and eels oriented to the southeast. This orientation was lost during ebb tide, when the moon was below the horizon. During full moon and third quarter, we did not observe patterns in orientation behaviour.”

L158 Regarding, “The orientation of glass eels in our study was related to the moon’s azimuth and it’s position relative to the horizon”, yes it was statistically related in 3 out of 8 of the conditions, and only clearly in one case, so its hard to see how such a simple statement can be used, without more detailed wording.

We clearly and transparently state that the orientation of glass eels is related to the moon azimuth only at new moon and first quarter throughout the manuscript: in the abstract, in the results, in the figures, in the discussion. Therefore, we do not understand the concern of the reviewer with respect to our conclusions and interpretation of the data.

L183 Yes, so more detailed descriptions like this should also be in the Results and then mentioned generally like this in the Discussion.

In response to an earlier related comment from this reviewer we added a statement about the results during first quarter at the beginning of the discussion.

Line 152 :“Significant patterns in orientation direction were also observed during first quarter, but only during flood tide, when the moon was above the line of the horizon. However, the orientation response was 75° from the moon azimuth and oriented to the southeast, and it was lost during ebb tide, when the moon was below the horizon.”

L187 New moon flood seems no more clear than first quarter flood, so why only mention new moon. i.e. the statistical result either means something or it does not. The clear result is new moon ebb obviously.

The analysis in Figure 4 shows that glass eels significantly orient towards the moon azimuth at new moon during both ebb and flood tides (Fig. 4C column iii). Thus, this behavior does not depend on the tidal phase.

I don't see the logic of saying “only when the moon was invisible to them” as support for the mechanism cant be visual. The tests were in daytime, so the moon was always invisible to them, and they are under water, so they will never see the moon regardless of drifting at day or night. At night they can see “moonlight” but not the moon itself in most cases. Please clarify if what you are talking about, such as during your testing, or in general unrelated to the testing periods.

The moon was invisible to them because the percentage of moon visible to the eye was 0-10% during new moon even if the moon was above the horizon, and it was also daytime. We edited the sentence as follows. “The behaviour of the glass eels showed the strongest lunar-related patterns in orientation when the moon was invisible to them (at new moon during daytime)”.

L206 clarify the increase is compared to what?

L 217: we added “baseline value occurring during daytime, when the moon is still below the horizon”

L232 need to add the reference number again and not just the author names.

Reference added.

L243 A single sentence paragraph is appropriate? ...could be play a role... is a typo

We fixed this so that there is no longer a one sentence paragraph.

L263 I am losing the logic here since you found no trend during full moon. And the concept of

swimming south also needs clarifying, since the text is saying that is a mean direction of the moon across the whole time the moon is up. So the eels start out swimming east, then SE, S, SW and then almost west following the moon direction, resulting in a mean south direction? That seems strange and maladapted. This raises the question then about why you don't discuss the time of day of your tests? That would affect where the moon (and sun) is in the sky.

We discuss the possibility that glass eels could orient (on average) to the south also during full moon nights because it is the only other period of the lunar cycle when the global-scale lunar disturbance in electrical fields increases. We describe this in detail in the previous section of the discussion. With regards to the direction; the reviewer is correct and that is why we always specify, throughout the manuscript, that the southerly direction is the average direction. We conducted tests throughout the day during all the moon phases, which provide data throughout almost the whole range of moon azimuth. We did not perform tests at night, which we state and address in the manuscript and suggest as work that needs to be conducted in future studies.

L267 Recruitment at new and full moon is also thought to be related to strong tidal flows isn't it?
-species of glass eels- not eel

L 279: We added "when the strongest tidal flows also occur."

Figures

Figure 2: Fonts range from readable to microscopic. Including for the same y-axis label of "Latitude".

We increased the font size of the axes of Figure2

The size range issue applies to almost all figures, but there is also obvious variability of use of bold text, types of fonts, use of capital first letters in most of the figures.

We have now made the font consistent in all figures.

Figure 5 has a technical issue of the flows of the Gulf Stream being incorrect, or bad even for a schematic drawing. The most obvious is current flowing over what appears to be the continental shelf off Atlantic Canada. However, unless I am really missing something here, I would suggest that this figure is not justified to include, and crosses beyond what your data show.

We modified the diagram of the Atlantic gyre so that it is further from Atlantic Canada. We argue for the retention of this Figure because it is essential to explain why the lunar-related orientation could be ecologically relevant for glass eels and we think that it will help to guide the readers through the discussion.

A few examples of what to check for in all Supplementary figures:
Figure S2 gives the units, but not what it is.

We added “swimming speed” to the Y axis.

Figure S3 has much larger font than S2

The font is now the same size.

Figure S4 has bold x-axis labels, the y-axis does not

The label was made bold on purpose, as it refers to all of the categories of the x axis, which are not in bold.

Figure S8 lower case y-axis label, upper case x-axis

We have now made the font size consistent